# Certified Circuits: Stability Guarantees for Mechanistic Circuits

**Alaa Anani** [1 2]   **Tobias Lorenz** [2]   **Bernt Schiele** [1]   **Mario Fritz** [* 2]   **Jonas Fischer** [* 1]

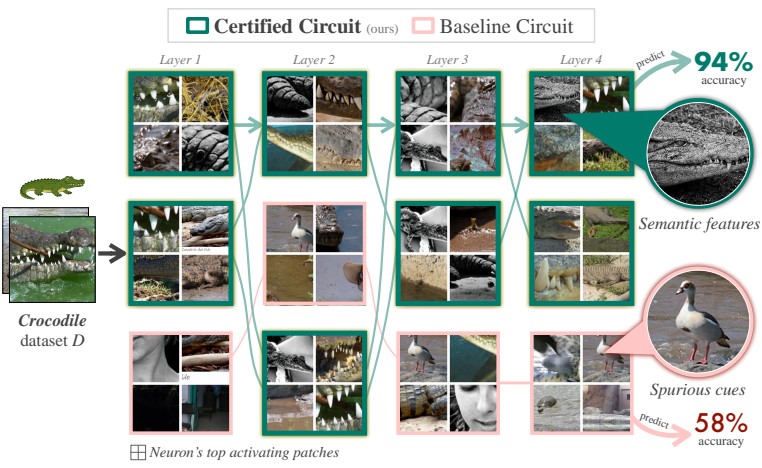

**(a)** Certified circuits, here for 'African crocodile', remove spurious cues.

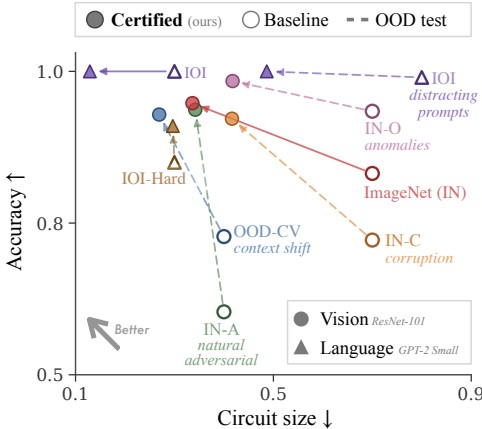

**(b)** Certified circuits are more compact (x) *and* more accurate (y).

*Figure 1.* **Certified circuits are smaller, more accurate, and generalize to OOD.** Given a concept dataset, we isolate a circuit—a subnetwork encoding that concept—that is provably stable under dataset edits. (a) Certified circuits keep stable semantic neurons (e.g., teeth) and abstain from unstable spurious ones (e.g., bird), improving the 'African crocodile' circuit accuracy to $94\%$. (b) Across distribution shifts (OOD test), certified circuits are significantly smaller and more accurate than baseline circuits.

## Abstract

Understanding *how* neural networks arrive at their predictions is essential for debugging, auditing, and deployment. Mechanistic interpretability pursues this goal by identifying *circuits*—minimal subnetworks responsible for specific behaviors. However, existing circuit discovery methods are brittle: circuits depend strongly on the chosen concept dataset and often fail to transfer out-of-distribution, raising doubts whether they capture the concept or merely dataset-specific artifacts. We introduce *Certified Circuits*, which provide provable stability guarantees for circuit discovery. Our framework wraps any black-box discovery algorithm with randomized data subsampling to certify that inclusion decisions over cir-

cuit components—neurons or edges of the model graph, depending on the base algorithm—are invariant to bounded edit-distance perturbations of the concept dataset. Unstable components are abstained from, yielding circuits that are more compact and more accurate. We validate across three architectures (ResNet, ViT, GPT-2) on vision (ImageNet and four OOD datasets) and language (IOI, IOI-Hard, Greater-Than) tasks. Certified circuits achieve up to 56% higher accuracy and up to 80% fewer components, and remain reliable where baselines degrade. *Certified Circuits* puts circuit discovery on formal ground by producing mechanistic explanations that are provably stable and better aligned with the target concept. Code: https://github.com/AlaaAnani/certified-circuits.

*Equal contribution  [1]Max Planck Institute for Informatics, Saarland Informatics Campus, Saarbrücken, Germany [2]CISPA Helmholtz Center for Information Security, Saarbrücken, Germany. Correspondence to: Alaa Anani <aanani@mpi-inf.mpg.de>.

*Proceedings of the $43^{rd}$ International Conference on Machine Learning*, Seoul, South Korea. PMLR 306, 2026. Copyright 2026 by the author(s).

## 1. Introduction

Understanding *how* neural networks arrive at their predictions is a central challenge in machine learning. Mechanistic interpretability tackles this by identifying *circuits*—minimal subnetworks responsible for specific model behav-

iors (Olah et al., 2020; Conmy et al., 2023; Elhage et al., 2021). In vision models, for instance, a circuit for recognizing "crocodile" might comprise particular convolutional filters detecting scales, sharp teeth, and elongated snouts, connected through specific neurons across subsequent layers. Discovering such circuits promises not only scientific insight into learned representations but also practical benefits: debugging failure modes, auditing for biases, and enabling targeted model editing.

Circuit discovery methods have emerged across modalities. In language models, activation patching and causal tracing isolate attention heads and MLP neurons responsible for factual recall, indirect object identification, and other behaviors (Meng et al., 2022; Wang et al., 2023; Goldowsky-Dill et al., 2023). In vision, analogous methods prune model components to find minimal sufficient subnetworks for recognizing visual concepts (Rajaram et al., 2023; Olah et al., 2020; Dreyer et al., 2024; Żukowska et al., 2026). These approaches start with a *concept dataset*: a collection of inputs representing the target behavior. They then identify which model components are necessary or sufficient to maintain performance on this dataset, discarding the rest. The result is a sparse circuit, i.e., a subgraph of the model graph intended to capture the mechanistic basis of the behavior.

However, current circuit discovery methods lack robustness (Méloux et al., 2025; uit de Bos & Garriga-Alonso, 2024; Miller et al., 2024; Friedman et al., 2024). The identified circuits are sensitive to the choice of concept dataset: adding, removing, or substituting a few semantically equivalent examples can change the discovered circuit unpredictably. They also fail to generalize to out-of-distribution (OOD) data. A circuit found using photographs of crocodiles on land may perform poorly on crocodiles in water, cartoon crocodiles, or crocodiles from unusual angles. Both issues stem from the same underlying problem—current methods overfit to the particular concept dataset rather than recovering the actual concept representation. This undermines confidence in the mechanistic explanation these methods produce.

To address this, we introduce *Certified Circuits* (Fig. 2), a framework that computes the first dataset-level robustness guarantees for circuit discovery. Given a concept dataset $\mathcal{D}$ and any black-box circuit discovery algorithm, we construct a certified circuit $C^*$ with the following guarantee:

> For any concept dataset $\mathcal{D}'$ within edit distance $r$ of $\mathcal{D}$, the certified circuit $C^*$ provably remains unchanged.

Edit distance counts insertions, deletions, or substitutions of examples—so $r = 5$ guarantees stability under any combination of up to five such changes. This covers an exponentially large family of datasets.

Our framework is *algorithm-agnostic*: we (i) randomly subsample the concept dataset several times, (ii) run the base algorithm on each subsample to obtain candidate circuits, and (iii) aggregate votes for each individual circuit component to determine which components are guaranteed to represent the concept across perturbed datasets. A key byproduct is *adaptive sparsity*: certification identifies components for which no robust decision can be made, yielding circuits that are more compact and accurate than fixed top-$K$ baselines.

In summary, we make the following **contributions**:

1. We introduce *Certified Circuits*, the first framework providing provable, algorithm-agnostic robustness guarantees for circuit discovery.
2. We derive **provable bounds** on certified radii and characterize their dependence on a probability threshold and deletion probability required for subsampling.
3. We demonstrate empirically across three architectures (ResNet, ViT, GPT-2), two modalities (vision, language) and four discovery algorithms that *Certified Circuits* are **more compact**, **more sufficient**, and **generalize better to OOD** than uncertified baselines while being **highly structurally stable**.

We validate Certified Circuits across two modalities and three architectures, evaluating *sufficiency* (does the circuit preserve model behavior?) and *compactness* (how small can the circuit be while sufficient?). For vision, we test ResNet-50/101 and ViT-B/16 on ImageNet and four OOD benchmarks (OOD-CV, ImageNet-A, ImageNet-O, ImageNet-C) with neuron-level top-$K$ scoring (relevance, activation, rank). For language, we test GPT-2 Small on IOI, IOI-Hard, and Greater-Than with EAP-IG (Hanna et al., 2024) as the base edge-level algorithm. Certified circuits consistently outperform uncertified baselines: certified accuracy improves by up to 56% on OOD shifts using up to 41% fewer neurons, and language circuits use up to 80% fewer edges at matched or higher accuracy. These gains hold across modalities, architectures, and base algorithms, confirming that certification captures more transferable mechanistic structure while pruning components that are not robustly necessary (Fig. 1). We further analyze structural convergence when circuits are rediscovered on shifted distributions and across random seeds. Certified Circuits lift classic circuit-based mechanistic explanations to *provably* robust and more compact explanations that empirically better generalize to OOD data.

## 2. Related works

### 2.1. Mechanistic Interpretability

Mechanistic interpretability aims to reverse-engineer the internal computations of neural networks, moving beyond input-output behavior to understand *how* models arrive at

their predictions (Olah et al., 2020; Elhage et al., 2021; Bereska & Gavves, 2024). The field spans observational methods that analyze learned representations (e.g., probing, sparse autoencoders) and interventional methods that causally localize computations through targeted ablations and activation patching (Zeiler & Fergus, 2014; Zimmermann et al., 2021; Meng et al., 2022).

**Circuit discovery.** *Circuit discovery* aims to identify minimal subnetworks (*circuits*) that implement a target behavior or concept. Circuit components—nodes and edges of the computational graph—can correspond to feature channels in CNNs, attention heads or head-to-head connections in transformers, MLP neurons, etc., depending on the chosen granularity (Bereska & Gavves, 2024). Typically, one (i) defines a concept via a dataset, (ii) represents the model as a computational graph with nodes and edges connecting them, and (iii) extracts a sparse subgraph whose components are necessary and/or sufficient for that behavior (Conmy et al., 2023). In language models, this approach has revealed circuits for factual recall (Meng et al., 2022), indirect object identification (Wang et al., 2023), and most recently for verifying chain-of-thought reasoning (Zhao et al., 2026). ACDC (Conmy et al., 2023) automates discovery via iterative edge pruning. In vision, circuits have been studied via feature-preserving subnetworks (Hamblin et al., 2022), connectivity-based tracing of concept-specific computations (Rajaram et al., 2023; Wang et al., 2019), disentanglement of polysemantic neurons into concept circuits (Dreyer et al., 2024), and qualitative connectome-style visualizations spanning all layers (Kowal et al., 2024).

**Stability limitations.** A core limitation is that discovered circuits can be *highly unstable*: swapping in different (but semantically equivalent) examples to represent the same concept (or making small additions/removals) can yield substantially different circuits. This raises a basic question: *is the circuit capturing the concept, or overfitting to dataset-specific spurious cues?* Recent work documents and diagnoses such fragility. Méloux et al. (2025) cast circuit discovery as statistical estimation and show that EAP-IG (a transformer circuit discovery method) circuits change markedly even when the *same behavior* is defined using paraphrased prompts, indicating high variance in the discovered structure. Miller et al. (2024) further find that common circuit faithfulness metrics are not robust to evaluation choices. Finally, Friedman et al. (2024) show that explanations can appear faithful on the discovery distribution yet fail to generalize, creating *interpretability illusions*.

These works highlight the problem but do not provide solutions that rule out instability. Concurrent work uses neural network verification to certify the faithfulness of a *fixed* circuit under small *input-level* perturbations (Hadad et al., 2026); in contrast, our concern is *dataset-to-circuit*

instability, where the *discovered circuit itself* changes when the concept dataset changes. To our knowledge, no prior method certifies invariance of circuit structure under bounded dataset-level perturbations.

## 2.2. Robustness Certification

A *robustness certificate* is a worst-case guarantee of *output invariance*: given an input $x$ and a radius $r$, the certified model provably returns the same prediction for every perturbed input $x'$ with $\mathrm{dist}(x, x') \le r$ (otherwise it abstains). Randomized smoothing yields such certificates by aggregating the base model's predictions under random perturbations and converting the resulting probability margin into a certified radius (Lécuyer et al., 2019; Cohen et al., 2019; Anani et al., 2025). Beyond classification, smoothing has been extended to (i) *structured outputs* via per-component abstention, certifying only confident components in segmentation (Fischer et al., 2021; Anani et al., 2024), and (ii) *discrete objects* under edit distance, where RS-Del uses randomized deletions to certify invariance against insertions, deletions, and substitutions within an edit budget (Huang et al., 2023). We build on these ideas to certify circuit component stability under *dataset*-level edit perturbations.

**Summary.** Circuit discovery is unstable: small edits to the concept dataset, even replacing examples with semantically equivalent ones, can produce entirely different circuits, blurring whether the circuit encodes the concept or dataset-specific artifacts. We address this by certifying the *dataset-to-circuit* mapping: using deletion-based smoothing (RS-Del) (Huang et al., 2023) to model bounded dataset edits, together with circuit component-level certification to exclude the unstable circuit components (Fischer et al., 2021), we return *certified circuits* whose certified components are provably invariant to edits within an edit distance.

## 3. Certified Circuits

To overcome the circuit instability in prior work, we formalize circuit discovery as a dataset-level mapping and ask: which circuit components are *provably stable* under bounded edits to the concept dataset? Our approach is driven by three goals. First, we seek guarantees (not empirical heuristics) that circuit structure is invariant, ruling out brittleness by construction. Second, edit distance provides a threat model: it captures the scenario where a practitioner adds examples, removes outliers, or swaps in semantically equivalent images, and expects the discovered circuit to remain unchanged if it encodes the concept. Third, stability under such edits is a prerequisite for *out-of-distribution generalization*: a circuit that changes when the concept dataset is perturbed cannot be expected to transfer to shifted test distributions. Enumerating all bounded-edit datasets is in-

feasible, so we use randomized smoothing: we run circuit discovery on many random subsamples and aggregate the outcomes to certify stability for all edits within the radius.

**Our approach (Figure 2).** Given a model graph $G=(\mathcal{V},\mathcal{E})$, a concept dataset $\mathcal{D}$, and any black-box circuit discovery algorithm $A$, we certify which circuit components (vertices $v \in \mathcal{V}$ or edges $e \in \mathcal{E}$) are provably stable under bounded dataset edits. After briefly reviewing randomized smoothing, we formalize our setup (§3.1). We then define *dataset deletion smoothing* (RS-Del (Huang et al., 2023); §3.2) and a *smoothed circuit discovery* rule that aggregates component-wise inclusion probabilities and thresholds them to output *certified in/out* or *abstain* (§3.3). We present Theorem 3.1, which guarantees certified decisions are invariant for all datasets within edit-distance $r$. Finally, we describe the Monte Carlo estimation of the certified circuit discovery algorithm (§3.4) and summarize its key properties (§3.5).

**Background: Randomized Smoothing**

We introduce randomized smoothing (Lécuyer et al., 2019; Cohen et al., 2019) as the main technical tool for turning empirical stability under random perturbations into *certified* robustness guarantees. Let $f_b : \mathcal{X} \to \mathcal{Y}$ be a base classifier and let $\phi : \mathcal{X} \to D(\mathcal{X})$ be a perturbation mechanism that maps an input $x$ to a distribution $\phi(x)$ over perturbed inputs. Randomized smoothing defines the smoothed classifier

$$f(x) := \arg\max_{y \in \mathcal{Y}} \mathbb{P}_{z \sim \phi(x)}[f_b(z) = y]. \quad (1)$$

A certificate is obtained by lower-bounding the probability of the predicted class and deriving a radius $r$ such that, with confidence at least $1 - \alpha$, the prediction is invariant within the corresponding neighborhood, i.e., $f(x) = f(x')$ for all $x'$ satisfying $\mathrm{dist}(x, x') \leq r$.

**3.1. Setup and Inputs**

Let $G = (\mathcal{V}, \mathcal{E})$ denote the model as a directed computational graph, where vertices $\mathcal{V}$ are computation units (e.g., neurons, attention heads) and edges $\mathcal{E}$ represent connections between them. A circuit $C$ is a subgraph of $G$ specified by selecting some subset of components. Let $\mathcal{U}$ denote the set of *circuit components* over which the base algorithm operates: $\mathcal{U} \subseteq \mathcal{V}$ for node-level methods (e.g., neuron-level top-$K$) and $\mathcal{U} \subseteq \mathcal{E}$ for edge-level methods (e.g., EAP-IG). We additionally define a *concept dataset* $\mathcal{D} = (x_1, \ldots, x_{|\mathcal{D}|}) \in \mathcal{X}^*$, viewed as a finite sequence of inputs that contain the same concept (e.g., same-class images).

Let black-box *circuit discovery algorithm* $A$ map a concept dataset to a binary mask over circuit components:

$$A : \mathcal{X}^* \longrightarrow \{0, 1\}^{|\mathcal{U}|}, \quad (2)$$

where $A_u(\mathcal{D}) = 1$ indicates that component $u \in \mathcal{U}$ is included in the circuit associated with $\mathcal{D}$ and $A_u(\mathcal{D}) = 0$ indicates exclusion. Two common examples of $A$ are: (i) node-level top-$K$, which scores vertices on $\mathcal{D}$ (e.g., by mean activation, gradient-based relevance, or per-example rank) and retains the top fraction per layer as sparse mask over $v \in \mathcal{V}$ (Olah et al., 2020; Hamblin et al., 2022; Rajaram et al., 2023; Dreyer et al., 2024); and (ii) edge-level attribution (e.g., EAP-IG (Hanna et al., 2024)), which scores edges of the computational graph via integrated gradients and retains the top-scoring fraction, yielding a sparse mask over $e \in \mathcal{E}$.

The goal is to construct a *smoothed* (certified) variant $\tilde{A}^\tau$ of $A$ whose component-wise decisions are provably stable under bounded edit perturbations of $\mathcal{D}$.

**3.2. Dataset Deletion Smoothing**

To certify robustness of circuit discovery under dataset-level edits, we model the concept dataset $\mathcal{D} = (x_1, \ldots, x_{|\mathcal{D}|})$ as a sequence and measure perturbations via edit distance $\mathrm{dist}_{\mathrm{edit}}(\mathcal{D}, \mathcal{D}')$, counting insertions, deletions, and substitutions required to transform $\mathcal{D}$ to $\mathcal{D}'$. Checking stability under all edits is intractable; instead, following RS-Del (Huang et al., 2023), we use *randomized deletions* as a smoothing perturbation, which yields certificates with respect to the full edit distance. Although concept datasets are naturally unordered, we fix an arbitrary ordering solely to define $\mathrm{dist}_{\mathrm{edit}}$; this does not affect the certificate since our base algorithms $A$ depend only on permutation-invariant statistics.

We define a deletion-based perturbation distribution $\phi_{p_{\mathrm{del}}}(\mathcal{D})$ by sampling an i.i.d. binary mask

$$\varepsilon = (\varepsilon_1, \ldots, \varepsilon_{|\mathcal{D}|}), \qquad \varepsilon_i \sim \mathrm{Bernoulli}(1 - p_{\mathrm{del}}),$$

where $\varepsilon_i = 1$ keeps $x_i$ and $\varepsilon_i = 0$ deletes it. This produces an *ordered subsequence*:

$$\mathcal{D} \odot \varepsilon := (x_i \mid \varepsilon_i = 1)$$
$$= (x_{i_1}, \ldots, x_{i_m}), \quad \text{with } 1 \leq i_1 < \cdots < i_m \leq |\mathcal{D}|.$$

The random sub-dataset $\mathcal{D} \odot \varepsilon$ is distributed according to $\phi_{p_{\mathrm{del}}}(\mathcal{D})$. This perturbation model is the input-side mechanism underlying our certified guarantees, and it directly connects our setting to RS-Del's edit-distance certification under deletion-based smoothing (Huang et al., 2023).

**3.3. Smoothed Circuit Discovery**

Given a base circuit discovery algorithm $A$ and the deletion perturbation distribution $\phi_{p_{\mathrm{del}}}(\mathcal{D})$, we define the *smoothed* circuit discovery algorithm

$$\tilde{A}^\tau : \mathcal{X}^* \to \{0, 1, \oslash\}^{|\mathcal{U}|},$$

which returns for each component $u \in \mathcal{U}$ one of three outcomes: *certified in* (1), *certified out* (0), or *abstain* ($\oslash$).

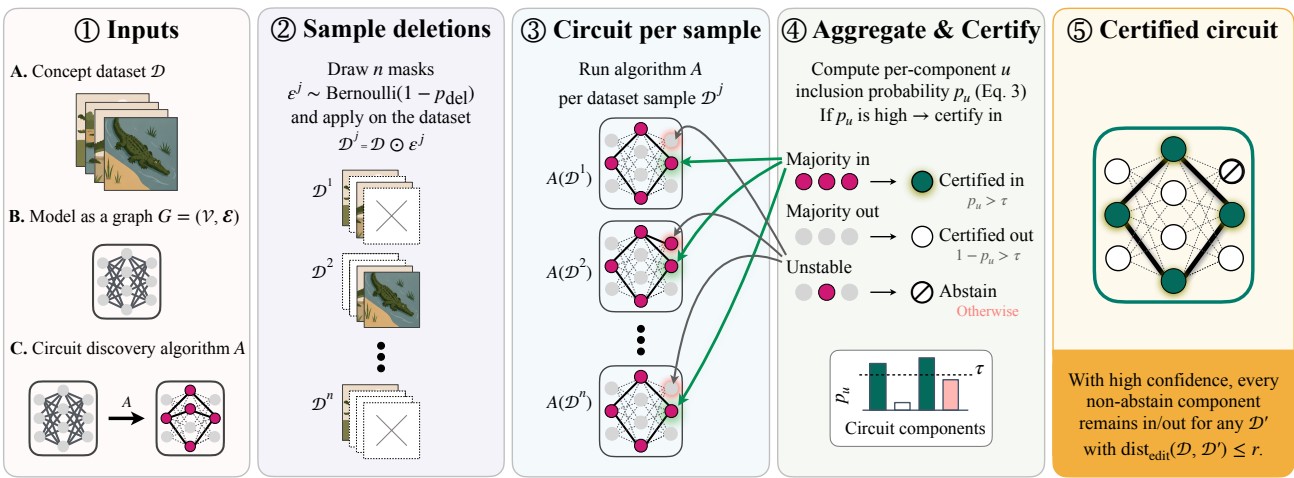

*Figure 2.* **Certified circuit discovery via concept deletion smoothing.** (§3.1) Given a concept dataset $\mathcal{D}$, model graph $G$, and circuit discovery algorithm $A$: (§3.2) We sample dataset variants via per-example deletion with probability $p_{\text{del}}$, (§3.3) run $A$ on each to obtain per-sample circuits, (§3.4) aggregate per-component (e.g., vertex) inclusion frequencies, (§3.4) certify components as *in*, *out*, or *abstain* ($\oslash$) based on votes consistency. (§3.5) The certified circuit is provably invariant to concept dataset edits within radius $r$.

The confidence threshold $\tau \in [0.5, 1)$ controls how much posterior mass is required to make a non-abstaining decision: if neither inclusion nor exclusion is sufficiently likely under randomized deletions, the method abstains.

To define $\tilde{A}^\tau$, we quantify how consistently algorithm $A$ includes a component $u$ under randomized deletions as the *smoothed inclusion probability*:

$$p_u(\mathcal{D}) := \mathbb{P}_\varepsilon\big[A_u\big(\mathcal{D} \odot \varepsilon\big) = 1\big],$$
$$\varepsilon_i \sim \text{Bernoulli}(1 - p_{\text{del}}) \text{ i.i.d.} \tag{3}$$

i.e., the probability that $A$ includes $u$ when run on a randomly deleted sub-dataset $\mathcal{D} \odot \varepsilon$. Values near $1$ mean $u$ is selected almost always (stable inclusion), while values near $0$ mean it is rarely selected (stable exclusion).

The smoothed algorithm $\tilde{A}^\tau$ converts these probabilities into *certified* per-component decisions by requiring a margin of at least $\tau$ in favor of inclusion or exclusion. Following segmentation-style smoothing (Fischer et al., 2021), we set

$$\tilde{A}_u^\tau(\mathcal{D}) = \begin{cases} 1 & \text{if } p_u(\mathcal{D}) > \tau, \\ 0 & \text{if } 1 - p_u(\mathcal{D}) > \tau, \\ \oslash & \text{otherwise,} \end{cases} \tag{4}$$

so $\tilde{A}_u^\tau(\mathcal{D}) = 1$ certifies that $u$ is robustly included, $\tilde{A}_u^\tau(\mathcal{D}) = 0$ certifies that it is robustly excluded, and $\tilde{A}_u^\tau(\mathcal{D}) = \oslash$ abstains when neither decision has enough evidence. We write $\tilde{A}^\tau(\mathcal{D}) := (\tilde{A}_u^\tau(\mathcal{D}))_{u \in \mathcal{U}} \in \{0, 1, \oslash\}^{|\mathcal{U}|}$ for the resulting three-valued mask over all components.

**Guarantees.** The rule defining $\tilde{A}^\tau$ converts vote consistency under randomized deletions into a worst-case guarantee over *all* datasets within an edit-distance neighborhood of $\mathcal{D}$.

Combining RS-Del (Huang et al., 2023) with component-wise certification (Fischer et al., 2021) yields a certified stability guarantee for every non-abstaining component:

---

**Theorem 3.1: Certified Circuit Robustness**

With confidence at least $1 - \alpha$, for any circuit component $u \in \mathcal{U}$ with $\tilde{A}_u^\tau(\mathcal{D}) \in \{0, 1\}$ and any perturbed dataset $\mathcal{D}'$ satisfying $\text{dist}_{\text{edit}}(\mathcal{D}, \mathcal{D}') \leq r$, the membership decision is invariant:

$$\tilde{A}_u^\tau(\mathcal{D}') = \tilde{A}_u^\tau(\mathcal{D}),$$

where the certified radius is

$$r := \left\lfloor \frac{\log(1.5 - \tau)}{\log p_{\text{del}}} \right\rfloor. \tag{5}$$

---

The guarantee in Theorem 3.1 states that any component that $\tilde{A}^\tau$ certifies as *in* (1) or *out* (0) of the circuit remains so for all concept datasets $\mathcal{D}'$ with $\text{dist}_{\text{edit}}(\mathcal{D}, \mathcal{D}') \leq r$, with confidence at least $1 - \alpha$. Components assigned $\oslash$ are excluded from the certified circuit. The proof is in App. A.

**From certified mask to a circuit.** Given the per-component guarantee in Theorem 3.1, we define the certified circuit as the subgraph of $G$ induced by certified-in components. Let

$$\mathcal{U}^* := \{u \in \mathcal{U} \mid \tilde{A}_u^\tau(\mathcal{D}) = 1\} \tag{6}$$

denote the set of certified-in components. The certified circuit $C^* = (\mathcal{V}^*, \mathcal{E}^*)$ is then defined as: (i) For *node-level methods* ($\mathcal{U} \subseteq \mathcal{V}$): $\mathcal{V}^* = \mathcal{U}^*$ and $\mathcal{E}^* = \{(v, w) \in \mathcal{E} \mid v, w \in \mathcal{V}^*\}$. (ii) For *edge-level methods* ($\mathcal{U} \subseteq \mathcal{E}$): $\mathcal{E}^* = \mathcal{U}^*$ and $\mathcal{V}^* = \bigcup_{(v,w) \in \mathcal{E}^*} \{v, w\}$. In both cases, components assigned $\oslash$ are excluded as they are not certifiably stable.

### 3.4. Estimating the Certified Circuit

In practice, the probabilities $p_u(\mathcal{D})$ (Eq. 3) are unknown and are estimated by Monte Carlo sampling: draw $n$ i.i.d. deletion masks $\varepsilon^{(1)}, \ldots, \varepsilon^{(n)} \sim \text{Bernoulli}(1 - p_{\text{del}})^{|\mathcal{D}|}$, evaluate $A_u(\mathcal{D} \odot \varepsilon^{(j)})$ for each $j$, and use the resulting empirical frequency to estimate the lower bound of $p_u(\mathcal{D})$, from which a $(1 - \alpha)$ lower confidence bound is computed to decide whether $\tilde{A}_u^\tau(\mathcal{D}) \in \{0, 1\}$ or $\oslash$. We follow the standard evaluation scheme as in (Fischer et al., 2021) and (Huang et al., 2023) in our estimation Algorithm 1 in App. C.3.

### 3.5. Properties of Certified Circuits

Smoothed circuit discovery yields certified circuits with three key properties: (i) **provable dataset-level stability**: all non-abstain certified components are invariant under any sequence of up to $r$ dataset edits; (ii) **spurious-feature suppression**: components whose membership decisions are unstable across deletions are abstained from, producing strictly sparser circuits (Fig. 1 (a), Fig. 5, App. F.6); and (iii) **algorithm and model agnosticism**: our framework wraps any circuit discovery method $A$ and model $G$ without requiring access to their internals. Next, we discuss that Certified Circuits also have *practical* benefits, better capturing the target concept prediction and generalizing to OOD data.

## 4. Experimental Setup

**Baseline circuit discovery algorithms.** We instantiate our certification framework with four standard base algorithms (Eq. 2). For *vision*, we follow the common top-$K$ flow (Hamblin et al., 2022; Conmy et al., 2023; Rajaram et al., 2023; Dreyer et al., 2024): candidate vertices $v \in \mathcal{V}$ are feature channels at each residual block's output, scored and ranked per layer, with top-$K$ fraction retained. We condsider three scorers: *relevance*, *activation*, and *rank* — using relevance by default in §5 and evaluating others in App. F.1. For *language*, we use EAP-IG (Hanna et al., 2024), which scores edges $e \in \mathcal{E}$ via integrated gradients and retains the top-$K$ fraction. For certified circuits, $K$ denotes the *effective* fraction of components retained after certification (averaged across layers) rather than the base algorithm's target $K$, since certification abstains on unstable components that the uncertified base algorithm would otherwise include. Thus, at a fixed $K$, the certified circuits effective size is always $\leq$ the baseline.

**Datasets and architectures.** For *vision*, we use ImageNet (Russakovsky et al., 2015) as the ID dataset, with each class defining a concept, and evaluate under four OOD shifts: OOD-CV (Zhao et al., 2022), ImageNet-A and ImageNet-O (Hendrycks et al., 2021), and ImageNet-C (Hendrycks & Dietterich, 2019). Circuits are discovered on 100 randomly selected classes. Main results use ResNet-101 and ResNet-

50 (He et al., 2016); ViT-B/16 results are in App. F.3. For *language*, we evaluate GPT-2 Small (Radford et al., 2019) on three tasks: indirect object identification (IOI) (Wang et al., 2023), IOI-Hard, and Greater-Than (Hanna et al., 2023). The task concept dataset is a prompt set; circuits are discovered on ID splits and evaluated on held-out ID and OOD splits. *Vision* and *language* setups are in App. B.1 & B.2.

**Sufficiency.** We measure circuit sufficiency by preservation of model predictions when computation is restricted to the discovered circuit, following standard practice (Conmy et al., 2023; Meng et al., 2022; Wang et al., 2023; Dreyer et al., 2024). For class-$c$ circuit $C_c$, we zero non-circuit channels at each residual block and evaluate on concept dataset $\mathcal{D}_c$. We report mean circuit-class accuracy

$$\text{cACC} := \frac{1}{|\mathcal{Y}|} \sum_{c \in \mathcal{Y}} \text{CA}(C_c, \mathcal{D}_c), \qquad (7)$$

where $\text{CA}(C_c, \mathcal{D}_c)$ is the pruned model's accuracy on $\mathcal{D}_c$ and $\mathcal{Y}$ is the set of evaluated classes. Language sufficiency is analogous, restricting computation to certified-in edges and measuring task accuracy on the prompt set.

**Certification hyperparameters.** The choice of $p_{\text{del}}$ and $\tau$ trades off certification strength against the base algorithm's operating conditions: high $p_{\text{del}}$ deletes more data, hindering the base algorithm, while high $\tau$ demands higher consistency, increasing the abstain rate. For *vision*, we default to $p_{\text{del}} = 0.6$ and $\tau = 0.95$ (certified radius $r = 1$) as a good sweet spot. For *language*, where larger concept datasets tolerate more aggressive deletion, we tune $p_{\text{del}}$ and $\tau$ per task and report the best configuration (App. Table. 2). Results at larger radii are in App. F.4 (*vision*) and E.2 (*language*). We use $n = 1{,}000$ Monte Carlo samples and failure probability $\alpha = 0.001$ throughout, following standard practice (Lécuyer et al., 2019; Cohen et al., 2019; Anani et al., 2025). The theoretical minimum $n$ is analyzed in App. C.2. All certified results hold with confidence $1 - \alpha = 99.9\%$.

## 5. Results

We evaluate *Certified Circuits* in four ways: **sufficiency and compactness** (§5.1), testing whether circuits preserve task performance when used in isolation and how small they can be; **feature visualization** (§5.2), qualitatively inspecting the features encoded by the circuits, **out-of-distribution generalization** (§5.3), measuring whether circuits discovered in-distribution retain accuracy under distribution shift; and **structural stability** (§5.4), assessing whether circuit structure remains stable when re-discovered on shifted data.

| Domain | Setting | Dataset / Task | cACC ↑ | | | | Size $K$ ↓ | | | |
| --- | --- | --- | --- | --- | --- | --- | --- | --- | --- | --- |
| | | | Full | Baseline | Certified | Δ | Full | Baseline | Certified | Δ |
| **Vision** ResNet-101 | ID | ImageNet | 0.78 | 0.83 | **0.95** | ↑14% | 1.000 | 0.700 | **0.336** | ↓52% |
| | OOD | ImageNet-A | 0.07 | 0.60 | **0.94** | ↑56% | 1.000 | 0.400 | **0.342** | ↓15% |
| | | OOD-CV | 0.20 | 0.73 | **0.93** | ↑28% | | 0.400 | **0.269** | ↓33% |
| | | ImageNet-C | 0.57 | 0.72 | **0.92** | ↑28% | | 0.700 | **0.416** | ↓41% |
| | | ImageNet-O | 0.81 | 0.93 | **0.98** | ↑6% | | 0.700 | **0.417** | ↓41% |
| **Language** GPT-2 Small | ID | IOI | 1.00 | 1.00 | 1.00 | 0% | 1.000 | 0.300 | **0.129** | ↓58% |
| | | IOI-Hard | 1.00 | 1.00 | 1.00 | 0% | | 0.200 | **0.125** | ↓38% |
| | | Greater-Than | 1.00 | 1.00 | 1.00 | 0% | | 0.040 | **0.010** | ↓75% |
| | OOD | IOI | 0.98 | 0.99 | **1.00** | ↑2% | 1.000 | 0.800 | **0.487** | ↓40% |
| | | IOI-Hard | 0.79 | 0.85 | **0.91** | ↑8% | | 0.300 | **0.297** | ↓2% |
| | | Greater-Than | 1.00 | 1.00 | 1.00 | 0% | | 0.040 | **0.008** | ↓80% |

*Table 1.* **Certified vs. baseline circuit sufficiency.** Peak cACC and corresponding circuit size $K$ under sufficiency pruning on ResNet-101 for vision datasets and GPT-2 Small for language tasks. Circuits are discovered on ID datasets and evaluated on ID or on OOD shifts. Δ is relative improvements on unrounded values. Certification hyperparameters are in App. Table 2. All three vision discovery algorithms in App. Table 5.

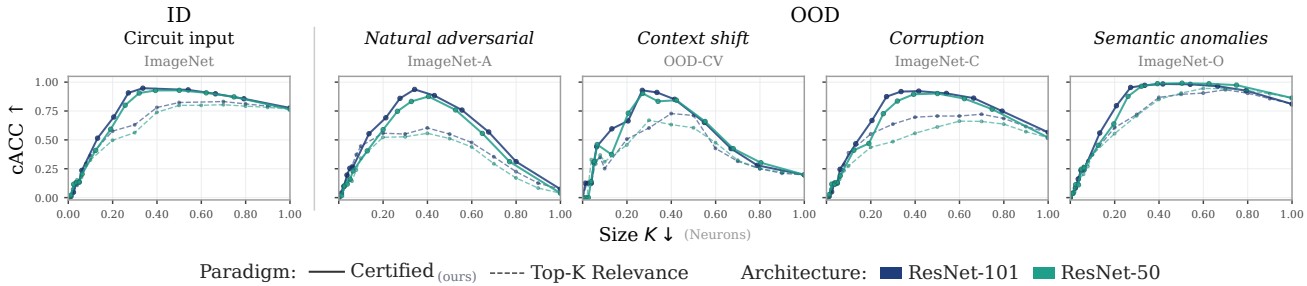

*Figure 3.* **Circuit accuracy (cACC) vs. size $K$.** Solid lines show certified circuits, dashed show the baseline, with colors distinguishing models. Circuits are discovered on ImageNet and evaluated on ID or OOD data. This figure is extended to ViT-B/16 in App. Fig. 12 and language circuits in App. Fig. 9 & 10.

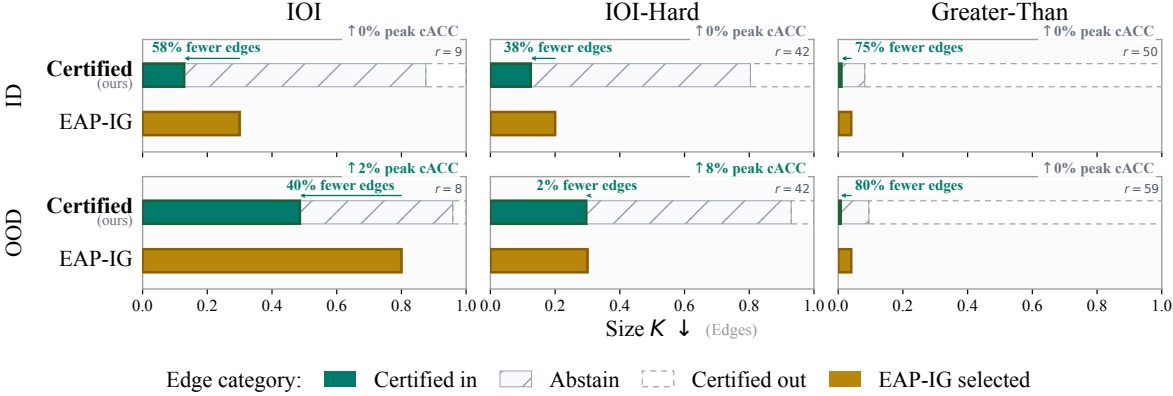

*Figure 4.* Edges of the **certified vs. baseline EAP-IG circuits on GPT-2 Small at peak cACC.** Annotations report the relative edge reduction and the gain in peak cACC of the certified circuit over EAP-IG. The certified radius $r$ denotes the best value for every task.

## 5.1. Circuit Sufficiency and Compactness

We study two questions: *(i) sufficiency*: does the circuit alone preserve the target class prediction? and *(ii) compactness*: how small can the circuit be while remaining sufficient? We sweep the circuit size $K \in (0, 1]$ (fraction of

components retained), and report (a) the *peak* cACC over $K$ and the corresponding $K$ (Table 1), (b) the cACC–$K$ curves on *vision* across five datasets (Fig. 3) and (c) the edge-level circuit breakdown on *language* tasks at peak cACC (Fig. 4).

**Sufficiency.** *Does using the circuit alone preserve the*

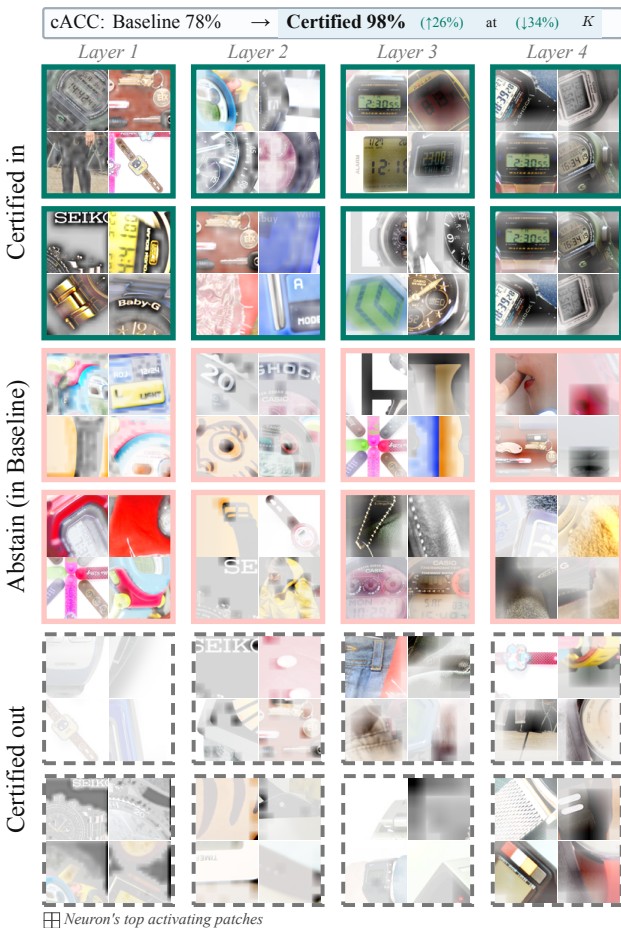

cACC: Baseline 78% → **Certified 98%** (↑26%) at (↓34%) K

*Layer 1*  *Layer 2*  *Layer 3*  *Layer 4*

Certified in

Abstain (in Baseline)

Certified out

⊞ *Neuron's top activating patches*

*Figure 5.* **Features encoded by certified-in, abstain (within Baseline), and certified-out neurons** (class: `digital watch`, ResNet-101). Per layer, top-activating neurons in each category.

*class/task accuracy?* Across *vision* and *language*, certified circuits match or exceed baseline peak cACC (and the full model) on all datasets and tasks (Table 1, Fig. 3). On ImageNet (ID), certified circuits reach 95% vs. the baseline's 83% (↑ 14%). On *language*, peak cACC is saturated at 100% for both methods on all three ID tasks.

**Compactness.** *How small can the circuit be while sufficient?* Across both modalities, certified circuits reach peak cACC at substantially smaller size $K$ than baselines (Table 1, Fig. 3 (*vision*), Fig. 4 (*language*)). On ImageNet, the certified circuit is 52% smaller while improving cACC by 14%, with similar reductions across other datasets. On *language*, where sufficiency is already saturated, compactness is the dominant signal: certified circuits match EAP-IG's peak cACC on ID while using 58%, 38%, and 75% fewer edges on IOI, IOI-Hard, and Greater-Than respectively (Fig. 4).

**Effect of compactness on sufficiency.** Fig 3 plots cACC against the circuit size $K$ across five vision datasets. Both certified and baseline circuits follow a three-stage pattern as $K$ increases: (i) small K yields insufficient circuits that omit class-critical neurons; (ii) intermediate K captures the necessary evidence and cACC peaks; (iii) large K introduces competing or spurious features, reducing class specificity and degrading cACC. Certified circuits shift this curve favorably in both dimensions, peaking at smaller $K$ (more compact) and at higher cACC (more sufficient), pruning components that are unnecessary and harmful, rather than trading one property for the other. On *language*, both curves plateau rather than degrade at large $K$; certified circuits plateau at higher/equal cACC at smaller $K$ (App. Fig. 9).

> Certified circuits are **substantially smaller**, **more accurate** on *vision* and **matching accuracy** on *language*.

### 5.2. Feature Visualization of Circuits

To inspect what certification removes, Fig. 5 visualizes per-layer neurons in three categories: certified-in, abstain (within the baseline circuit), and certified-out. Certified-in neurons consistently respond to class-defining features (watch faces, dials). Abstained neurons, included by the baseline but dropped by certification, predominantly fire on co-occurring but non-class-specific cues (hands, wrists) or on spurious patterns. Certified-out neurons rarely fire on the concept dataset at all. The resulting `digital watch` certified circuit improves cACC by 26% at 34% smaller size, suggesting that abstained components are not merely uninformative but actively harmful, biasing the baseline toward spurious class correlates. Additional classes are in App. F.6.

> Certified circuits **abstain from spurious cues**.

### 5.3. Out-of-Distribution Generalization

*Do circuits discovered in-distribution retain accuracy on shifted data?* Certified circuits transfer substantially better than baselines, with the largest gains on the hardest shifts (Table 1, Fig. 3 (*vision*), Fig. 4 (language)). On *vision*, ImageNet-discovered certified circuits reach 93% cACC on OOD-CV (↑ 28% over baseline) at 33% smaller $K$, and 94% on ImageNet-A (↑ 56%) at 15% smaller $K$—shifts where the full model collapses to 20% and 7%. On *language*, certified circuits match or improve EAP-IG's peak cACC under OOD prompts at 40%, 2%, and 80% fewer edges on IOI, IOI-Hard, and Greater-Than (Fig. 4), with up to ↑ 8% cACC on IOI-Hard. Results suggest that certified circuits capture concept relevant features that transfer across shifts. Abstention removes components with unstable inclusion under dataset perturbations, often spurious, yielding circuits that generalize beyond the discovery distribution.

> Certified circuits **generalize to OOD shifts**.

## 5.4. Structural Stability Under Distribution Shift

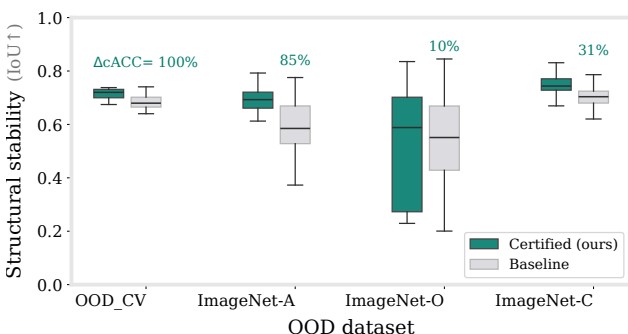

*Figure 6.* **Structural stability under distribution shift.** Per-class IoU between circuits discovered on ImageNet and re-discovered on each OOD dataset, at the $K$ maximizing the certified–baseline $\Delta$cACC gap. Boxes show the distribution over classes.

The previous experiments used the same circuit on different distributions. A stronger test asks: *does the certified circuit structure remain stable when re-discovered on shifted distributions of the same concept?* We measure this via IoU between circuits discovered on ImageNet and re-discovered on each shifted dataset (Fig. 6). For two circuits with vertex sets $V$ and $V'$, the IoU is $|V \cap V'|/|V \cup V'|$.

Overall, certified circuits have higher median IoU and tighter distributions than baselines across shifts, indicating more consistent structure. The largest stability gains align with the largest $\Delta$cACC performance gains on OOD-CV (100%) and ImageNet-A (85%). ImageNet-O is the exception, where both methods show high variance, suggesting class-dependent circuit reconfiguration under semantic anomalies.

These findings reinforce the OOD generalization results: abstention suppresses shift-sensitive vertices, concentrating circuits on an invariant core that better captures the underlying concept rather than dataset-specific artifacts.

> Certified circuits **converge to a similar invariant core** when re-discovered on shifted distributions.

## 6. Discussion and Limitations

**Discussion.** Certified circuits outperform baselines across three architectures (ResNet, ViT, GPT-2) and two modalities (vision, language). For sufficiency and compactness, certification shifts the accuracy–sparsity curve: circuits peak at smaller sizes with higher cACC, suggesting unstable components are not only unnecessary but harmful. Feature visualization confirms this: abstained neurons fire on co-occurring but non-class-specific cues, while certified-in neurons consistently respond to class-defining features. For OOD generalization, certified circuits discovered in-distribution (ImageNet) transfer without re-discovery, improving cACC by up to 56% on ImageNet-A while reducing

circuit size, and producing language circuits up to 80% smaller at matched or higher accuracy. Structural stability results support this mechanism: certified circuits show higher IoU when re-discovered on OOD data, indicating convergence to an invariant core. Beyond cross-distribution stability, certified circuits also converge across random seeds (App. F.2), confirming the certification procedure itself is reliable. Together, these findings support our hypothesis that the instability noted in prior work (Méloux et al., 2025; Miller et al., 2024; Friedman et al., 2024) arises from components inconsistently selected across dataset variants; abstention removes them, yielding circuits that provably reflect the concept rather than dataset-specific artifacts.

**Limitations.** Certifying a circuit requires running the discovery algorithm on $n$ randomized deletion masks, but caching forward and backward passes across masks keeps wall-clock cost close to running the base algorithm once ($\sim 2.4\times$ overhead at $n=1000$; App. D), so runtime does not linearly scale in $n$. Larger certified radii require higher deletion rates, which can degrade the base algorithm on very small concept datasets; however, certification recovers full performance at larger $|\mathcal{D}|$, supporting radii up to $r=59$ (App. F.4, E.2). We use a single sparsity $K$ across layers; layer-wise budgets inspired by pruning literature may yield finer sparsity allocation. Concurrent work (Hadad et al., 2026) certifies circuits via neural-network verification, but its guarantees are over continuous *input* perturbations to a *fixed* circuit, and its reliance on exact verifiers restricts it to small models and datasets (MNIST, CIFAR-10, GTSRB); our guarantees instead cover *dataset-level* edits to the *discovered* circuit, and scale to any standard vision and language models. While we expect the framework to extend to larger language models, multimodal models, and sparse-activation architectures, experimental validation remains open.

## 7. Conclusion

We introduced *Certified Circuits*, a framework providing provable stability guarantees for circuit discovery. By combining deletion-based randomized smoothing with per-circuit-component abstention, our method certifies that circuits remain unchanged under bounded dataset edits—directly addressing the instability undermining confidence in mechanistic explanations. Empirically, certified circuits are more compact, more sufficient, and generalize better to OOD data, while remaining structurally stable beyond their certified radius. Our work establishes that reliable circuit discovery is achievable: practitioners can obtain mechanistic explanations that are provably stable and empirically robust, bridging interpretability and trustworthiness.

## Impact Statement

This paper advances the field of machine learning by introducing Certified Circuits, a framework that provides formal stability guarantees for mechanistic circuit discovery methods. By enabling provably robust mechanistic explanations, our work strengthens the reliability and scientific validity of interpretability analyses, particularly under dataset variation and distribution shift.

We anticipate that this contribution will have positive downstream impacts in areas where trustworthy model understanding is critical, such as model debugging, robustness evaluation, and auditing of learned representations. More stable and transferable explanations may help practitioners better identify spurious correlations, understand failure modes, and design safer machine learning systems.

At the same time, as with interpretability tools more broadly, there is a risk that certified explanations could be misinterpreted as complete or definitive accounts of model behavior, despite capturing only a subset of the underlying computation. We emphasize that certified circuits provide guarantees relative to a specific threat model and concept dataset, and should be used as one component within a broader interpretability and evaluation toolkit.

Overall, we believe the societal implications of this work are aligned with established goals in machine learning interpretability—improving transparency, robustness, and trustworthiness—and do not raise new ethical concerns beyond those already present in the field.

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

# Appendix

## A. Proof of Theorem 3.1

*Proof.* Fix a component $u \in \mathcal{U}$. Define the per-component base classifier $h_u : \mathcal{X}^* \to \{0,1\}$ by $h_u(\mathcal{D}) := A_u(\mathcal{D})$. Let $\phi_{p_{\text{del}}}(\mathcal{D})$ denote the RS-Del perturbation that independently deletes each element of the sequence $\mathcal{D}$ with probability $p_{\text{del}}$, producing a random subsequence $\mathcal{D} \odot \varepsilon$.

**Smoothed probabilities and abstaining decision.** Define the smoothed inclusion probability

$$p_u(\mathcal{D}) := \Pr_{\varepsilon \sim \text{Bernoulli}(1-p_{\text{del}})^{|\mathcal{D}|}} [h_u(\mathcal{D} \odot \varepsilon) = 1],$$

and let $p_{u,0}(\mathcal{D}) := 1 - p_u(\mathcal{D})$ and $p_{u,1}(\mathcal{D}) := p_u(\mathcal{D})$. Let the (non-abstaining) smoothed label be

$$\tilde{A}_u(\mathcal{D}) := \arg \max_{c \in \{0,1\}} p_{u,c}(\mathcal{D}), \qquad \mu_u(\mathcal{D}) := \max\{p_u(\mathcal{D}), 1 - p_u(\mathcal{D})\}.$$

We output a *certified* (possibly partial) decision by thresholding as in segmentation-style smoothing with abstention:

$$\tilde{A}_u^\tau(\mathcal{D}) = \begin{cases} 1 & \text{if } p_u(\mathcal{D}) > \tau, \\ 0 & \text{if } 1 - p_u(\mathcal{D}) > \tau, \\ \oslash & \text{otherwise.} \end{cases}$$

Note that $\tau \geq \frac{1}{2}$ implies that whenever $\tilde{A}_u^\tau(\mathcal{D}) \in \{0,1\}$, the maximizer $\tilde{A}_u(\mathcal{D})$ is unique and equals $\tilde{A}_u^\tau(\mathcal{D})$.

**RS-Del certificate on the input sequence.** Apply RS-Del (Huang et al., 2023) to the smoothed binary classifier $\tilde{A}_u$ under Levenshtein edit distance (allowing insertions, deletions, and substitutions). RS-Del (Theorem 7 and Table 1 in (Huang et al., 2023)) gives that if the predicted class at $\mathcal{D}$ has confidence $\mu_u(\mathcal{D})$, then the smoothed prediction is invariant for any edit-distance ball of radius $r \leq r^\star(\mu_u(\mathcal{D}))$, where

$$r^\star(\mu) = \left\lfloor \frac{\log(1 + \nu(\eta) - \mu)}{\log p_{\text{del}}} \right\rfloor.$$

For binary outputs and symmetric thresholds (our case), $\nu(\eta) = \frac{1}{2}$, hence

$$r^\star(\mu) = \left\lfloor \frac{\log(1.5 - \mu)}{\log p_{\text{del}}} \right\rfloor.$$

Moreover $r^\star(\mu)$ is nondecreasing in $\mu$ because $\log p_{\text{del}} < 0$ and $1.5 - \mu$ decreases with $\mu$.

**Conclude invariance for certified vertices.** Assume $\tilde{A}_u^\tau(\mathcal{D}) \in \{0,1\}$. Then $\mu_u(\mathcal{D}) > \tau$, so by monotonicity $r^\star(\mu_u(\mathcal{D})) \geq r^\star(\tau)$. Define

$$r := r^\star(\tau) = \left\lfloor \frac{\log(1.5 - \tau)}{\log p_{\text{del}}} \right\rfloor.$$

Then for every $\mathcal{D}'$ with $\text{dist}_{\text{edit}}(\mathcal{D}, \mathcal{D}') \leq r$, RS-Del implies the smoothed label is invariant:

$$\tilde{A}_u(\mathcal{D}') = \tilde{A}_u(\mathcal{D}) = \tilde{A}_u^\tau(\mathcal{D}).$$

This is exactly the claimed circuit-membership invariance for all non-abstaining vertices.

**Statistical confidence.** In practice $\mu_u(\mathcal{D})$ is unknown and we certify based on a $(1 - \alpha)$ lower confidence bound (e.g., Clopper–Pearson) for $\mu_u(\mathcal{D})$. On the event that the true confidence exceeds this bound (which holds with probability at least $1 - \alpha$), the above argument applies. If one wants the guarantee to hold *simultaneously* for all vertices (Fischer et al., 2021), set per-vertex failure probability to $\alpha/|\mathcal{U}|$ and apply a union bound. $\square$

| Domain | | Setting | Dataset / Task | $\tau$ | $p_{\text{del}}$ | $r$ |
|---|---|---|---|---|---|---|
| **Vision** ResNet-101 | | ID | ImageNet | 0.95 | 0.60 | 1 |
| | | OOD | ImageNet-A | 0.95 | 0.60 | 1 |
| | | | OOD-CV | 0.95 | 0.60 | 1 |
| | | | ImageNet-C | 0.95 | 0.60 | 1 |
| | | | ImageNet-O | 0.95 | 0.60 | 1 |
| **Language** GPT-2 Small | | ID | IOI | 0.90 | 0.95 | 9 |
| | | | IOI-Hard | 0.85 | 0.99 | 42 |
| | | | Greater-Than | 0.90 | 0.99 | 50 |
| | | OOD | IOI | 0.85 | 0.95 | 8 |
| | | | IOI-Hard | 0.85 | 0.99 | 42 |
| | | | Greater-Than | 0.95 | 0.99 | 59 |

*Table 2.* **Certified algorithm configuration per dataset and task in Table 1.**

# B. Experimental Setup

Table 2 lists the certification hyperparameters ($\tau$, $p_{\text{del}}$, radius $r$) used for each result in Table 1.

## B.1. Vision Setup

**Concept datasets.** Each ImageNet class $c$ defines a concept dataset $\mathcal{D}_c$ as a set of same-class images. We use $|\mathcal{D}_c| = 50$ images per class from the ImageNet validation split. Circuits are discovered on 100 randomly selected classes and tested on ID (same validation images) and OOD evaluation sets.

**OOD evaluation datasets.** We evaluate ImageNet-discovered circuits under four out-of-distribution shifts, each targeting a distinct failure mode. **OOD-CV** (Zhao et al., 2022) places objects in novel contexts, backgrounds, poses, and weather conditions (*context shift*). **ImageNet-A** (Hendrycks et al., 2021) consists of natural adversarial examples mined to fool standard ImageNet classifiers (*natural adversarial*). **ImageNet-O** (Hendrycks et al., 2021) contains images of object categories *not* present in the ImageNet-1k label set (*semantic anomalies*). **ImageNet-C** (Hendrycks & Dietterich, 2019) applies synthetic corruptions to ImageNet images, and we pick *defocus blur* at severity level 2 (*corruption*).

For each OOD dataset we use the same 100 classes as on ID, except OOD-CV which only contains 6 classes.

**Architectures and vertex selection.** Main results use ResNet-101 and ResNet-50 (He et al., 2016), ImageNet-pretrained from `torchvision`. Circuit vertices are the feature channels at the output of each of the four residual blocks, giving 256, 512, 1024, and 2048 candidate channels per block respectively, for a total of 3,840 vertices on ResNet-101. The top-$K$ fraction is applied per layer. Transformer results use ImageNet-pretrained ViT-B/16 from `torchvision`. We instrument all 12 transformer encoder blocks. For block $i$, vertices are the 768 embedding-channel outputs of the second linear layer in the MLP sub-block, `encoder.layers[i].mlp[3]` (our hook `blocki_mlp_fc2`), before the subsequent dropout and residual addition. Token positions are not treated as separate vertices; each vertex corresponds to one MLP-output channel in one block. Thus ViT-B/16 has $12 \times 768 = 9,216$ candidate vertices. The top-$K$ fraction is applied independently within each block.

## B.2. Language Setup

**Model and tasks.** We evaluate GPT-2 Small (Radford et al., 2019) via HuggingFace (Wolf et al., 2020) on three next-token binary prediction tasks. Each example provides a clean prompt, a corrupted prompt, an answer token, and a distractor. IOI (Wang et al., 2023) predicts the indirect object in a two-name sentence, with corruption randomizing names while preserving the answer/distractor pair. IOI-Hard applies the same rule with longer prompts and harder OOD distractor clauses. For Greater-Than (Hanna et al., 2023), adapted to the EAP setting (Conmy et al., 2023; Syed et al., 2023), the clean prompt sets up a year span whose immediate successor is the answer, and corruption shifts the interval so the distractor becomes the natural continuation.

**Data.** We generate 100 training, 100 ID test, and 100 OOD test examples per task. Circuits are discovered on the training split and evaluated on held-out ID and OOD prompts. OOD splits are task-specific: object/place distractors (IOI), repeated Q&A detours (IOI-Hard), and pre-answer distractor insertion (Greater-Than). Table 3 shows example ID and OOD prompts.

| | ID | OOD (ours) |
|---|---|---|
| **IOI** | *Then, Jessica and Rachel went to the garden. Rachel gave a drink to* (Jessica / Rachel) | *Then in the morning, Jennifer and Nicholas went to the school. While Nicholas looked around the school, Jennifer talked about the snack, and after a while Nicholas gave a snack to* (Jennifer / Nicholas) |
| **IOI-Hard** | *Then in the morning, Nicole and Katherine went to the restaurant. Katherine gave a ring to* (Nicole / Katherine) | *Then in the morning, Elizabeth and Allison went to the office. Elizabeth repeatedly asked Allison about the necklace . . . and resumed the discussion of the necklace, Allison gave a necklace to* (Elizabeth / Allison) |
| **Greater-Than** | *In scenario 920499 at the school with the drink, The war lasted from the year 1799 to the year* (1800 / 1840) | *In scenario 794799 at the station with the necklace, The war lasted from the year 1399 to the year after a long and distracting discussion at the school about the basketball, followed by more unrelated details,* (1400 / 1440) |

*Table 3.* **OOD prompt construction for the three language tasks.** ID prompts follow the original task formulations; OOD prompts (ours) introduce task-specific distractors. Each cell shows the prompt followed by (answer / distractor).

**Circuit graph and metric.** EAP-IG operates on edges of the EAP computational graph (Syed et al., 2023). Nodes are the token embedding, every attention-head and MLP output, and the final logits. Candidate edges connect each earlier node to every downstream attention-head Q/K/V input, MLP input, and the final logits, yielding 32,491 candidate edges $u \in \mathcal{U}$. Circuit size $K$ is the retained edge fraction, and cACC is exact next-token accuracy (correct iff the argmax under the circuit-restricted computation equals the answer token).

**Certification hyperparameters.** We sweep $\tau \in \{0.85, 0.90, 0.95\}$ and $p_{\text{del}} \in \{0.85, 0.95, 0.99\}$. For each task/split, Fig. 9 reports the configuration with the highest peak cACC (ties broken by smaller effective size) while Fig. 10 fixes $r \in \{42, 50, 59\}$ and shows the cACC-$K$ curve across the three tasks.

## C. Certification Hyperparameter Theoretical Analysis

### C.1. Certified Radius $r$ vs. $p_{\text{del}}$ and $\tau$

The certified radius $r$ in Theorem 3.1 is determined by the deletion probability $p_{\text{del}}$ and confidence threshold $\tau$ (Eq. 5).

Figure 7 shows the certified radius $r$ at varying $p_{\text{del}}$ and $\tau$ values. The certified radius increases with both $p_{\text{del}}$ and $\tau$: higher deletion probabilities mean the smoothed algorithm aggregates over more aggressively perturbed sub-datasets, while higher confidence thresholds require stronger agreement across perturbations before certifying a decision. As $p_{\text{del}} \rightarrow 1$, the radius grows rapidly, but this comes at a cost: the base algorithm $A$ receives increasingly sparse sub-datasets, which can degrade its performance when concept datasets are small. Conversely, lower $p_{\text{del}}$ values preserve more examples per run but yield smaller certified radii.

In practice, we select $p_{\text{del}}$ to balance two considerations: (i) achieving a meaningful certified radius (e.g., $r \geq 1$ edits), and (ii) retaining enough examples per sub-dataset for the base algorithm to produce reliable circuits. For concept datasets of size $|\mathcal{D}|$, the expected sub-dataset size is $(1 - p_{\text{del}}) \cdot |\mathcal{D}|$, so larger concept datasets permit higher deletion probabilities without starving the base algorithm.

### C.2. Sample Complexity

Beyond the certified radius (App. C.1, Fig. 7), a second hyperparameter controlling the strength of our guarantees is the Monte Carlo sample budget $n$, which determines how tightly the empirical inclusion frequencies bound the true probabilities $p_u(D)$, which in turn determines whether a component reaches the threshold $\tau$ for certification. Fig. 8 reports the theoretical minimum $n$ required to certify a circuit under perfect agreement, as a function of $\tau$, with Bonferroni correction over

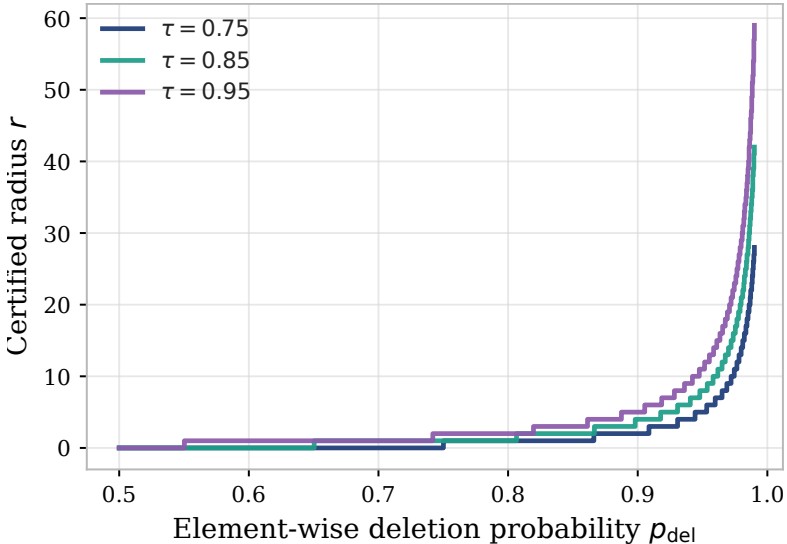

*Figure 7.* **Certified radius** $r$ **at various deletion probability** $p_{\text{del}}$ **values for different confidence thresholds** $\tau$. Larger $p_{\text{del}}$ and larger $\tau$ yield larger certified radii.

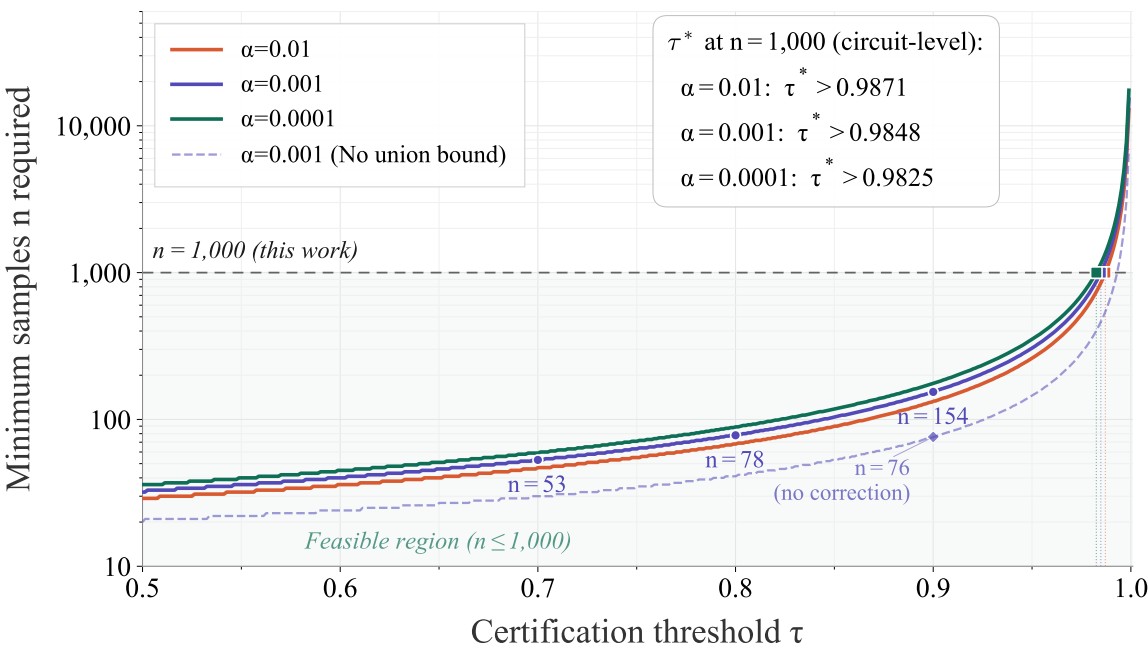

*Figure 8.* **Sample complexity vs. certification threshold.** Minimum samples $n$ required to certify a circuit at threshold $\tau$ under perfect agreement and Bonferroni correction over $N = 3,840$ components (the vertex set $\mathcal{V}$ of ResNet-101). $\tau^*$ is the largest threshold certifiable at a given $n$; any $\tau \le \tau^*$ is certifiable, so larger $n$ raises $\tau^*$. Solid curves: circuit-level guarantees at $\alpha \in \{0.01, 0.001, 0.0001\}$. Dashed: per-vertex (no correction).

$N = 3,840$ components (the number of candidate neurons in ResNet-101).

The required $n$ is small across the entire operating range: certification needs only 53–154 samples for $\tau \in [0.7, 0.9]$, and the union bound costs just $\approx 2\times$ over uncorrected per-vertex certification despite covering 3,840 simultaneous decisions. Sampling is therefore not a bottleneck. In practice, the budget of $n = 1,000$ used throughout the paper is conservative since it certifies any $\tau$ up to $\tau^* = 0.985$ at $\alpha = 0.001$, and could be reduced an order of magnitude at lower $\tau$ thresholds which produce a meaningful radius $r$.

### C.3. Estimating the Certified Circuit

Since we cannot invoke our theoretically constructed smoothed algorithm $\tilde{A}_u^\tau$ (Eq. 4) directly, we estimate it by Monte Carlo sampling in Algorithm 1. Our estimator is a novel combination of two randomized-smoothing frameworks, unifying them for the first time to certify circuit discovery: we adopt the *randomized-deletion sampling* of RS-Del (Huang et al., 2023), which perturbs a dataset by dropping examples i.i.d. and yields a certificate in *edit distance*, and the *component-wise output certification* of SEGCERTIFY (Fischer et al., 2021), which certifies each component of a structured output independently via a per-component abstention test. The key observation enabling this combination is that certifying circuit components is an instance of *binary segmentation*: each component $u \in \mathcal{U}$ plays the role of a pixel whose inclusion (1) or exclusion (0) must be certified independently, while the dataset $\mathcal{D}$ — rather than an image — is the object being perturbed. Casting circuit discovery this way lets us inherit SEGCERTIFY's certification algorithm while replacing its Gaussian input perturbation with RS-Del's dataset-level deletion, turning an $\ell_2$ certificate over pixels into an edit-distance certificate over datasets.

---

**Algorithm 1** CERTIFYCIRCUIT: estimating $\tilde{A}^\tau(\mathcal{D})$ (Eq. 4), combining SEGCERTIFY (Fischer et al., 2021) with RS-Del's deletion smoothing (Huang et al., 2023). Blue marks our changes.

> **function** CERTIFYCIRCUIT($A$, $p_{\text{del}}$, $\mathcal{D}$, $n$, $n_0$, $\tau$, $\alpha$)
> $\quad$ $\text{cnts}_1^0, \ldots, \text{cnts}_{|\mathcal{U}|}^0 \leftarrow$ SAMPLE($A$, $\mathcal{D}$, $n_0$, $p_{\text{del}}$)
> $\quad$ $\text{cnts}_1, \ldots, \text{cnts}_{|\mathcal{U}|} \leftarrow$ SAMPLE($A$, $\mathcal{D}$, $n$, $p_{\text{del}}$)
> $\quad$ **for** $u \in \mathcal{U}$:
> $\quad\quad$ $\hat{c}_u \leftarrow$ top index in $\text{cnts}_u^0$
> $\quad\quad$ $n_u \leftarrow \text{cnts}_u[\hat{c}_u]$
> $\quad\quad$ $\text{pv}_u \leftarrow$ BINPVALUE($n_u$, $n$, $\leq$, $\tau$)
> $\quad$ $r_1, \ldots, r_{|\mathcal{U}|} \leftarrow$ FWERCONTROL($\alpha$, $\text{pv}_1, \ldots, \text{pv}_{|\mathcal{U}|}$)
> $\quad$ **for** $u \in \mathcal{U}$:
> $\quad\quad$ **if** $\neg r_u$: $\hat{c}_u \leftarrow \oslash$
> $\quad$ $r \leftarrow \lfloor \log(1.5 - \tau) / \log p_{\text{del}} \rfloor$
> $\quad$ **return** $\hat{c}_1, \ldots, \hat{c}_{|\mathcal{U}|}, r$

---

**Algorithm 2** SAMPLE: per-component inclusion counts

> **function** SAMPLE($A$, $\mathcal{D}$, $n$, $p_{\text{del}}$)
> $\quad$ $\text{cnts}_u \leftarrow 0$ for all $u \in \mathcal{U}$
> $\quad$ **for** $j = 1, \ldots, n$:
> $\quad\quad$ $\varepsilon^{(j)} \sim \text{Bernoulli}(1 - p_{\text{del}})^{|\mathcal{D}|}$ $\qquad$ // RS-Del (Huang et al., 2023) deletion smoothing
> $\quad\quad$ $C^{(j)} \leftarrow A(\mathcal{D} \odot \varepsilon^{(j)})$
> $\quad\quad$ $\text{cnts}_u \leftarrow \text{cnts}_u + \mathbb{1}[u \in C^{(j)}]$ for all $u \in \mathcal{U}$
> $\quad$ **return** $\text{cnts} = (\text{cnts}_u)_{u \in \mathcal{U}}$

---

CERTIFYCIRCUIT (Algorithm 1) proceeds in two stages, each drawing samples through the primitive SAMPLE (Algorithm 2). SAMPLE draws $n$ deletion masks, runs the base algorithm $A$ on each resulting sub-dataset, and counts how often each component is included, giving the per-component $u$ inclusion counts $\text{cnts}_u$. The two stages use independent samples to avoid biasing the test by the choice of hypothesis. In the first stage, CERTIFYCIRCUIT draws $n_0$ *selection* samples and uses the counts to guess each component's likely decision $\hat{c}_u$ (include or exclude). In the second stage, it draws $n$ fresh *certification* samples and tests that guess: it computes a $p$-value $\text{pv}_u$ measuring how strongly the evidence supports $\hat{c}_u$ at confidence $\tau$. A component is certified if its $p$-value is small enough (via the FWER correction below) and abstains ($\oslash$) otherwise. Finally, CERTIFYCIRCUIT returns the edit-distance radius $r = \lfloor \log(1.5 - \tau) / \log p_{\text{del}} \rfloor$ (Eq. 5, from RS-Del), certifying every non-abstaining component against up to $r$ dataset edits.

**Controlling the family-wise error rate.** Because we certify $|\mathcal{U}|$ components simultaneously, the per-component failure probability $\alpha$ does not bound the probability that *any* certified decision is wrong; under the union bound this family-wise error rate (FWER) grows with $|\mathcal{U}|$. We therefore inherit the multiple-hypothesis-testing correction of Fischer et al. (2021): applying a Bonferroni correction over $\mathcal{U}$, we reject $H_0$ (and thus certify) only when $\rho_u \leq \alpha/|\mathcal{U}|$. This bounds the FWER at $\alpha$, so with confidence $1 - \alpha$ every non-abstaining component is certified correctly, and by Theorem 3.1 its membership is invariant for all $\mathcal{D}'$ with $\text{dist}_{\text{edit}}(\mathcal{D}, \mathcal{D}') \leq r$.

## D. Runtime Breakdown

| # Samples | Paradigm | ❶ Sample circuits Naive | Optimized | ❷ Decision rule | Total = ❶ + ❷ Naive | Optimized | Optimized speedup |
|---|---|---|---|---|---|---|---|
| $n = 1$ | Baseline | – | 1.007 s | – | – | 1.007 s | – |
| $n = 100$ | Majority vote | 101 s | 0.990 s | 0.006 s | 101 s | 0.996 s | ×101 |
| | Certified | 101 s | 0.990 s | 0.024 s | 101 s | 1.014 s | ×99.33 |
| $n = 500$ | Majority vote | 504 s | 1.516 s | 0.013 s | 504 s | 1.529 s | ×329 |
| | Certified | 504 s | 1.516 s | 0.102 s | 504 s | 1.618 s | ×311 |
| $n = 1000$ | Majority vote | 1007 s | 2.226 s | 0.021 s | 1007 s | 2.247 s | ×448 |
| | Certified | 1007 s | 2.226 s | 0.194 s | 1007 s | 2.420 s | ×416 |

*Table 4.* **Runtime breakdown for per-class ResNet-101 circuit discovery on the ImageNet validation set.** Naive scales linearly with $n$ while the optimized implementation caches per-image attribution score (e.g., relevance, activation) across samples. Hardware: Intel Core i7-14700 CPU, 62 GiB RAM, NVIDIA RTX 4090, batch size 50.

A potential concern with certified circuit discovery is sampling: running the base algorithm $n$ times, once per deletion mask, could scale runtime linearly in $n$, making the method impractical at larger $n$. Table 4 shows this concern does not materialize. We benchmark per-class circuit discovery on ResNet-101 over the ImageNet validation set, comparing a naive implementation (which re-runs the full discovery algorithm for each of the $n$ samples) against our optimized implementation that caches per-image scores (relevance) and computes per-sample circuits from the cache. Two findings emerge. First, naive scaling is indeed linear: $n = 1,000$ samples take $\approx 1,007$ s, which is $1,000\times$ the baseline single-circuit cost. Second, the optimized implementation reduces this to $2.42$ s, a $416\times$ speedup, because the dominant cost — forward and backward passes through the model — is shared across all $n$ samples. The decision rule itself (Eq. 4) takes under $0.2$ s even at $n = 1,000$. The optimized certified pipeline therefore runs in $\approx 2.4\times$ the cost of a single baseline circuit, independent of $n$ in practice.

# E. Additional Language Results

We extend on language circuit results illustrated earlier in §5.

## E.1. Certified Circuits at Best Radius $r$

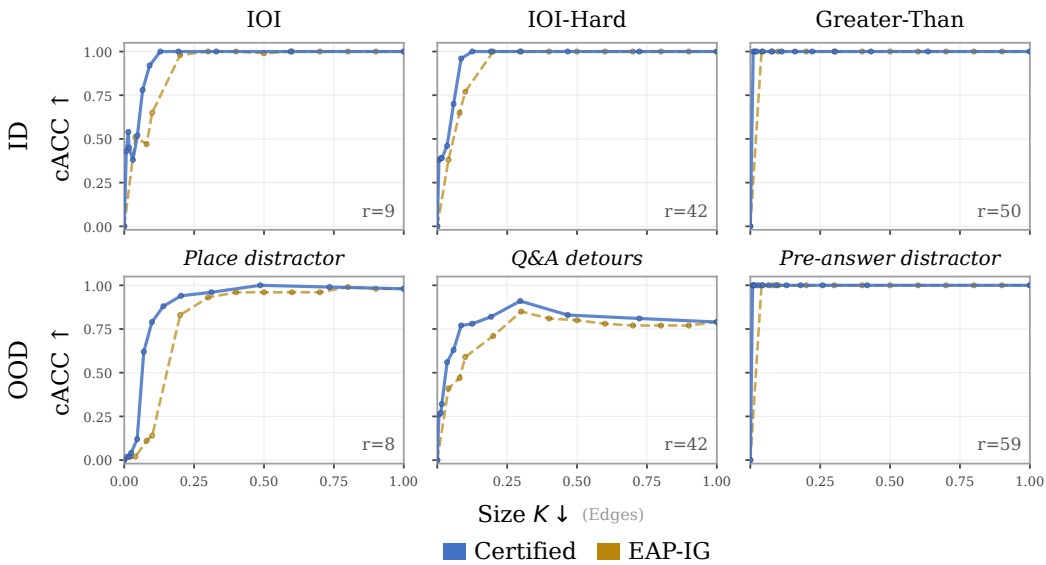

*Figure 9.* **Best certified vs. baseline EAP-IG circuits on GPT-2 Small.** cACC–$K$ curves for the certified circuit (highest peak cACC, blue solid) and EAP-IG (gold dashed) across three tasks (columns) and two settings (rows: ID, OOD). The certified radius $r$ for each circuit is annotated in the bottom-right of each panel.

Fig. 9 shows that on every task and setting, the certified curve reaches the EAP-IG peak at substantially smaller $K$, then remains saturated as $K$ grows. On IOI and Greater-Than, both ID and OOD curves are flat at $100\%$ from a small $K$ onward, with the certified curve reaching saturation 2–10× earlier than EAP-IG. IOI-Hard under Q&A detours is the only setting with a non-trivial peak: the certified circuit reaches $91\%$ cACC at $K \approx 0.3$, exceeding the EAP-IG peak ($85\%$) at matched size. Across all settings, certification matches or exceeds EAP-IG's peak cACC while certifying invariance to radii $r = 8$–$59$.

## E.2. Higher Certified Radii

Fig. 10 compares certified circuits at $r = 42$, 50, and 59 against EAP-IG on the same tasks. All three certified radii produce cACC–$K$ curves that are nearly indistinguishable from each other and that match or exceed EAP-IG at every $K$. On IOI and IOI-Hard, the three certified curves reach saturation at the same small $K$, with no measurable cACC penalty for the larger radius. On Greater-Than and Pre-answer distractor, the three curves overlap so tightly that the $r = 59$ certificate comes essentially for free: stability against up to 59 dataset edits is achieved with no loss in sufficiency. This confirms that, on tasks with sufficiently large concept datasets, the certified radius can be pushed well beyond the conservative $r = 1$ used for vision without degrading the discovered circuit.

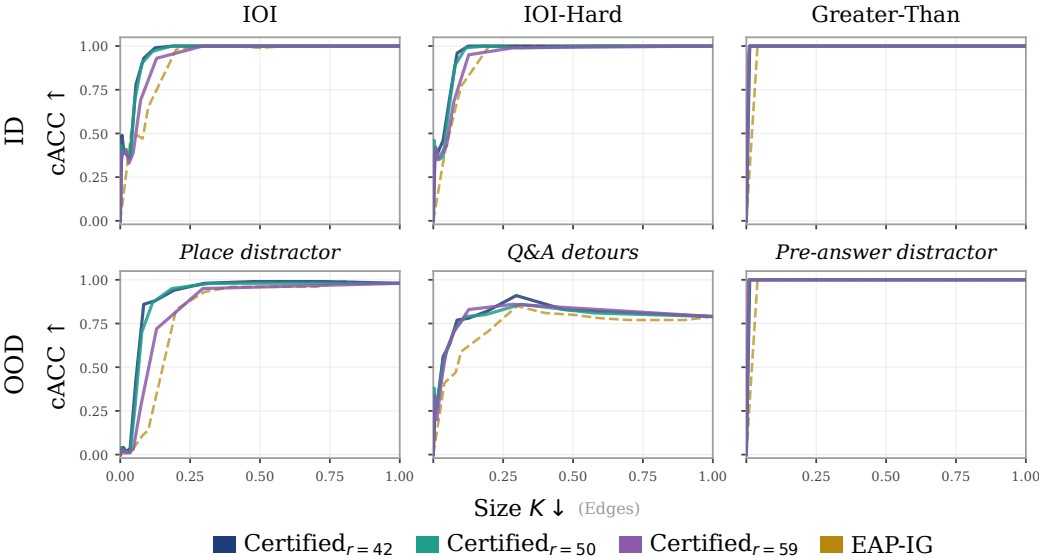

*Figure 10.* **Certified EAP-IG circuits at higher radii on GPT-2 Small.** cACC–$K$ curves for certified circuits at $r = 42$ (blue), $r = 50$ (teal), and $r = 59$ (purple), compared against EAP-IG (gold dashed) across three tasks (columns) and two settings (rows: ID, OOD). Curves overlap tightly, indicating that larger certified radii are achieved without measurable loss in cACC.

# F. Additional Vision Results

## F.1. Comparison of three circuit discovery algorithms

We extend the analysis in Table 1 from top-$K$ relevance to two additional scorers, *activation* and *rank*, on ResNet-101 (Table 5). The main results transfer across all three scorers: certified circuits are more accurate, smaller, and generalize better to OOD than their baselines. The largest cACC gains appear under OOD shifts: on ImageNet-A, certified circuits improve by $\uparrow 56\%$ (relevance), $\uparrow 84\%$ (activation), and $\uparrow 20\%$ (rank). Activation shows the largest relative gains because its uncertified baseline is weakest, while relevance reaches the highest absolute certified cACC (98% on ImageNet-O) and the largest compactness reductions ($\downarrow 52\%$ on ImageNet, $\downarrow 41\%$ on ImageNet-C and ImageNet-O). Rank yields smaller compactness reductions ($\downarrow 2\text{--}18\%$), consistent with rank-based scoring producing flatter score distributions that leave fewer components confidently unstable. We use relevance as the default in §5 based on its consistently higher absolute cACC and larger compactness gains.

| | | | cACC $\uparrow$ | | | | Size $K \downarrow$ | | | |
|---|---|---|---|---|---|---|---|---|---|---|
| Setting | Dataset | Top-$K$ | Full | Baseline | Certified | $\Delta$ | Full | Baseline | Certified | $\Delta$ |
| ID | ImageNet | Relevance | | $0.83 \pm 0.02$ | $\mathbf{0.95 \pm 0.02}^{\S}$ | $\uparrow 14\%$ | | $0.700$ | $\mathbf{0.336}$ | $\downarrow 52\%$ |
| | | Activation | $0.78 \pm 0.02$ | $0.80 \pm 0.02$ | $\mathbf{0.82 \pm 0.02}^{\S}$ | $\uparrow 3\%$ | $1.000$ | $0.700$ | $\mathbf{0.502}$ | $\downarrow 29\%$ |
| | | Rank | | $0.81 \pm 0.02$ | $\mathbf{0.82 \pm 0.02}^{\ddagger}$ | $\uparrow 2\%$ | | $0.700$ | $\mathbf{0.585}$ | $\downarrow 17\%$ |
| OOD | ImageNet-A | Relevance | | $0.60 \pm 0.03$ | $\mathbf{0.94 \pm 0.01}^{\S}$ | $\uparrow 56\%$ | | $0.400$ | $\mathbf{0.342}$ | $\downarrow 15\%$ |
| | | Activation | $0.07 \pm 0.01$ | $0.20 \pm 0.02$ | $\mathbf{0.36 \pm 0.03}^{\S}$ | $\uparrow 84\%$ | $1.000$ | $0.400$ | $\mathbf{0.330}$ | $\downarrow 18\%$ |
| | | Rank | | $0.24 \pm 0.02$ | $\mathbf{0.28 \pm 0.02}^{\S}$ | $\uparrow 20\%$ | | $0.600$ | $\mathbf{0.588}$ | $\downarrow 2\%$ |
| | OOD-CV | Relevance | | $0.73 \pm 0.11$ | $\mathbf{0.93 \pm 0.03}^{\dagger}$ | $\uparrow 28\%$ | | $0.400$ | $\mathbf{0.269}$ | $\downarrow 33\%$ |
| | | Activation | $0.20 \pm 0.09$ | $0.36 \pm 0.12$ | $\mathbf{0.54 \pm 0.13}^{\dagger}$ | $\uparrow 51\%$ | $1.000$ | $0.400$ | $\mathbf{0.268}$ | $\downarrow 33\%$ |
| | | Rank | | $0.44 \pm 0.13$ | $\mathbf{0.46 \pm 0.12}$ | $\uparrow 5\%$ | | $0.400$ | $\mathbf{0.394}$ | $\downarrow 2\%$ |
| | ImageNet-C | Relevance | | $0.72 \pm 0.02$ | $\mathbf{0.92 \pm 0.02}^{\S}$ | $\uparrow 28\%$ | | $0.700$ | $\mathbf{0.416}$ | $\downarrow 41\%$ |
| | | Activation | $0.57 \pm 0.02$ | $0.66 \pm 0.02$ | $\mathbf{0.71 \pm 0.02}^{\S}$ | $\uparrow 8\%$ | $1.000$ | $0.700$ | $\mathbf{0.502}$ | $\downarrow 29\%$ |
| | | Rank | | $0.67 \pm 0.02$ | $\mathbf{0.69 \pm 0.02}^{\ddagger}$ | $\uparrow 3\%$ | | $0.700$ | $\mathbf{0.585}$ | $\downarrow 17\%$ |
| | ImageNet-O | Relevance | | $0.93 \pm 0.01$ | $\mathbf{0.98 \pm 0.01}^{\S}$ | $\uparrow 6\%$ | | $0.700$ | $\mathbf{0.417}$ | $\downarrow 41\%$ |
| | | Activation | $0.81 \pm 0.01$ | $0.87 \pm 0.01$ | $\mathbf{0.90 \pm 0.01}^{\dagger}$ | $\uparrow 3\%$ | $1.000$ | $0.700$ | $\mathbf{0.492}$ | $\downarrow 30\%$ |
| | | Rank | | $0.88 \pm 0.01$ | $\mathbf{0.89 \pm 0.01}^{\dagger}$ | $\uparrow 2\%$ | | $0.700$ | $\mathbf{0.581}$ | $\downarrow 18\%$ |

*Table 5.* **Certified vs. baseline circuit sufficiency across three Top-K scoring algorithms with uncertainty estimates.** Peak cACC and corresponding circuit size $K$ under sufficiency pruning on ResNet-101. Circuits are discovered on ImageNet and evaluated either in-distribution or on OOD shifts. Significance markers compare certified circuits against the corresponding baseline circuit using paired class-level tests: $^{\dagger}p < 0.05$, $^{\ddagger}p < 0.01$, $^{\S}p < 0.001$.

## F.2. Stability Across Random Seeds

Recent work has raised concerns that mechanistic circuits are unstable across random seeds: Méloux et al. (2025) cast circuit discovery as statistical estimation and showed that EAP-IG circuits can vary substantially across runs even with the same concept dataset, indicating high variance in the discovered structure. This calls into question whether a single discovered circuit is meaningful, or merely one realization of a noisy procedure.

We investigate the stability of certified circuits across random seeds on ImageNet (Fig. 11). We run the certification pipeline ($n = 1,000$ deletion masks) 40 times with independent seeds on 100 ImageNet classes, and report all pairwise IoUs per class. Even on the 10 least stable classes, pairwise IoU stays above 0.95, with mean IoU $0.973 \pm 0.0015$ (95% CI) across all 100 classes. Certification is stable by construction: components are included only when their inclusion probability $p_u(D)$ exceeds the threshold $\tau$, so noise that would flip a borderline inclusion under a fixed top-$K$ rule instead resolves to abstention. Certification stabilizes circuits not only against bounded dataset edits (Theorem 3.1), but also empirically against the seed-level instability documented in prior work.

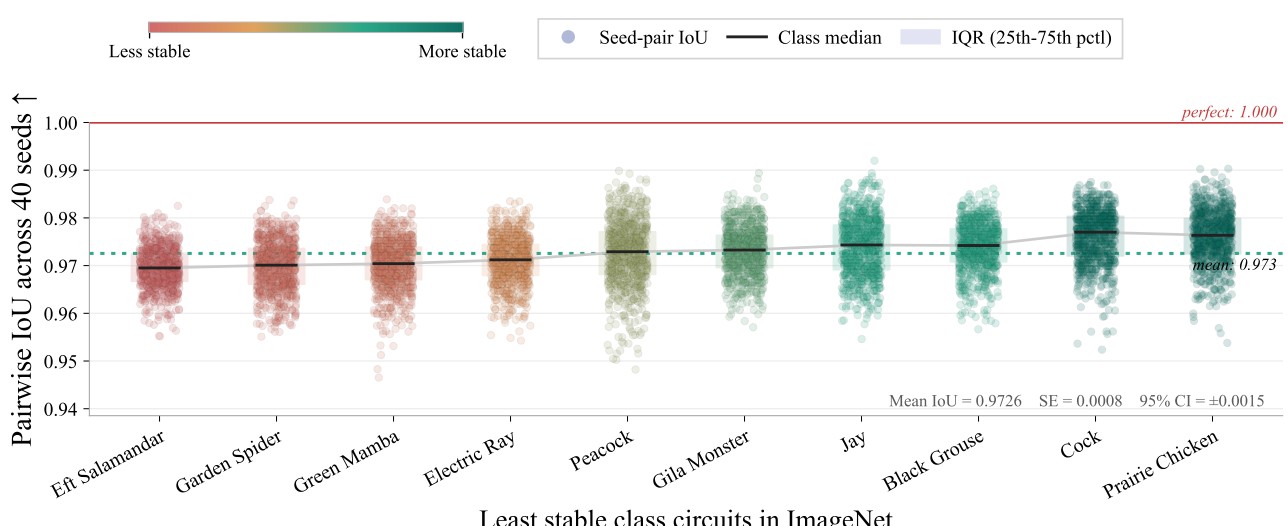

*Figure 11.* **Certified circuits are highly stable across seeds, even for the least stable classes.** Pairwise IoU between independent runs of the certification pipeline ($n$=1000 masks) across 40 seeds, for the 10 least stable ImageNet classes (ordered least to most stable); each dot is one seed-pair. Even in the worst case IoU exceeds 0.95. The dashed line marks the mean over 100 classes, $0.973 \pm 0.0015$ (95% CI).

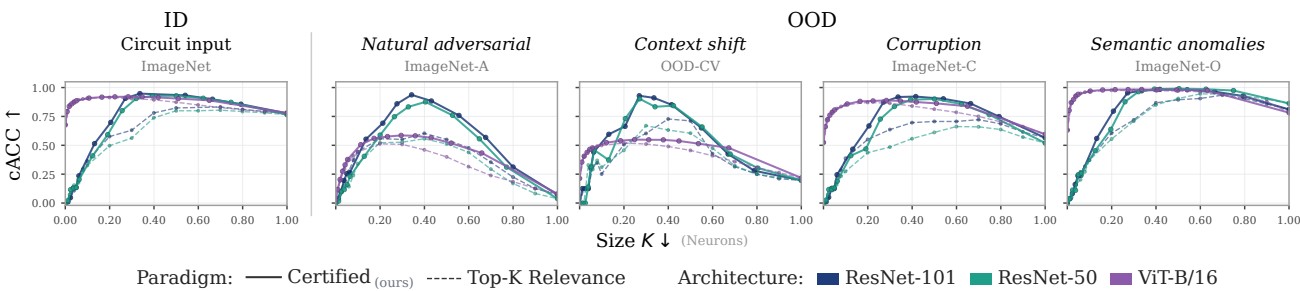

*Figure 12.* **Circuit accuracy (cACC) vs. size $K$ on ResNets and ViT-B/16.** Solid lines show certified circuits, dashed show the baseline, with colors distinguishing models. Circuits are discovered on ImageNet and evaluated either ID or on OOD data.

### F.3. Extending circuits to ViT-B/16 architecture

Fig. 12 extends the cACC–$K$ curves of Fig. 3 to ViT-B/16 (Dosovitskiy et al., 2021), alongside ResNet-101 and ResNet-50. Across all five datasets, certified circuits on ViT-B/16 closely track baseline top-$K$ relevance: peak cACC differs by at most ~2 points, and both methods peak at small $K$. The largest certified gains appear on ImageNet-A and OOD-CV, where certification recovers ~2 points of peak cACC at substantially smaller $K$ than the baseline, mirroring the trend observed on ResNets and confirming that the framework remains effective on vision Transformer architectures.

### F.4. Certified Vision Circuits at Larger Radii

In the main reported results §5, we use a conservative certified radius $r = 1$ for vision, which guarantees stability against a single dataset edit. Whether the framework can certify larger radii on vision depends on the relationship between the deletion probability $p_{\text{del}}$, the confidence threshold $\tau$, and the concept dataset size $|\mathcal{D}_c|$. Increasing $r$ at a fixed $\tau$ requires increasing $p_{\text{del}}$ (Eq. 5), which removes more examples per sample and can starve the base algorithm on small concept datasets. We characterize this trade-off across three experiments.

**Scaling to larger radii with larger $|\mathcal{D}_c|$.** Fig. 13 sweeps $r \in \{1, 3, 11, 59\}$ across $|\mathcal{D}_c| \in \{50, 100, 500, 1000\}$ on all five vision benchmarks. At $|\mathcal{D}_c| = 50$, larger radii ($r = 11, 59$) degrade noticeably because the base algorithm receives sub-datasets of only ~0–3 images per sample after deletion, too sparse to recover stable circuits. This degradation disappears as $|\mathcal{D}_c|$ grows: by $|\mathcal{D}_c| = 1,000$, certified circuits at $r = 59$ outperform the baseline on every dataset. The framework is not intrinsically limited to small radii on vision; the limitation is concept dataset size, and increasing it directly enables larger

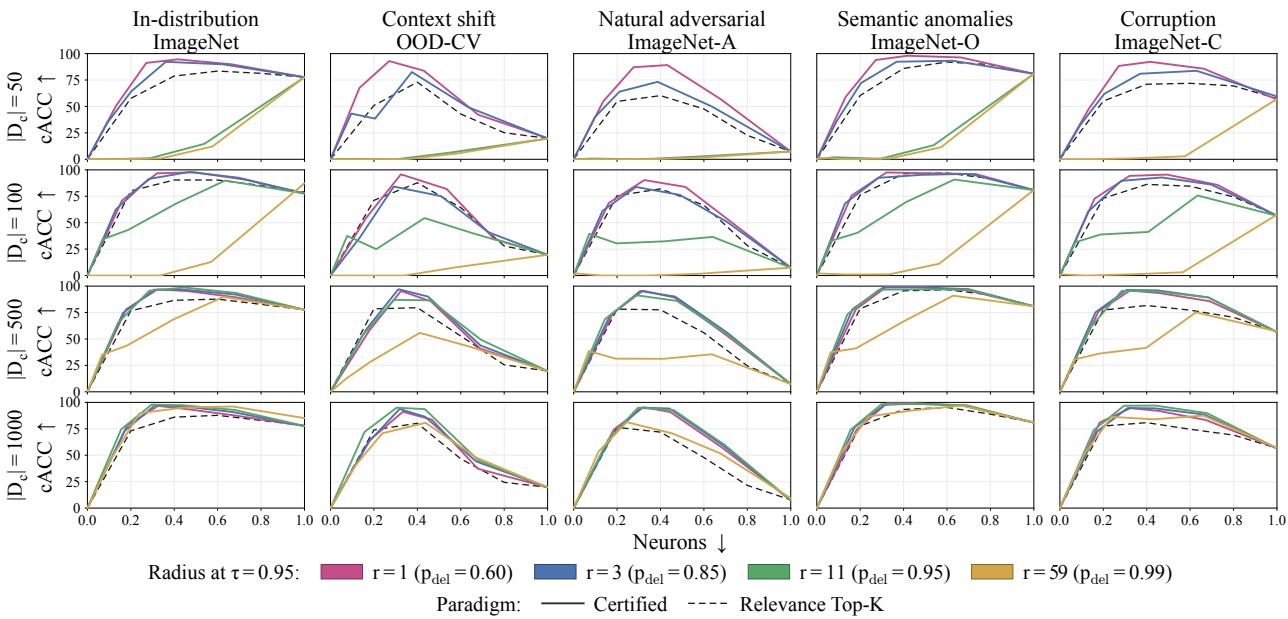

*Figure 13.* **Certified circuits scale to large radii given sufficient concept data.** cACC vs. circuit size for $r \in \{1, 3, 11, 59\}$ (colors) and $|\mathcal{D}_c| \in \{50, 100, 500, 1000\}$ (rows), across five ID/OOD benchmarks (columns). Solid: certified; dashed: baseline. Large radii degrade at small $|\mathcal{D}_c|$ but recover by $|\mathcal{D}_c| = 1,000$. ResNet-101, $\tau = 0.95$, $n = 1,000$.

certified radii.

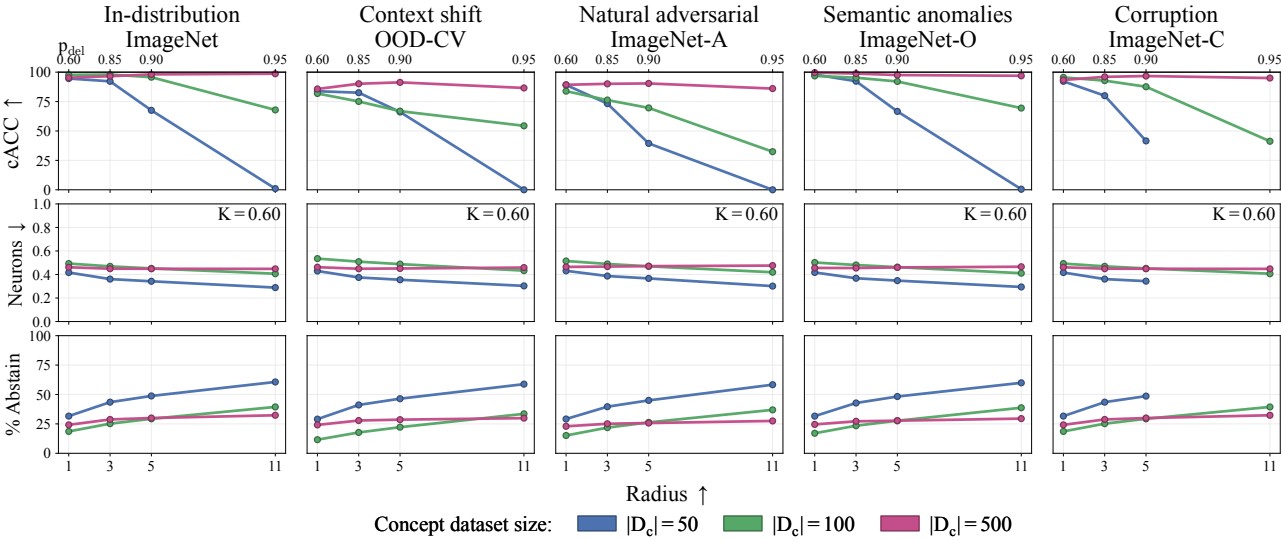

*Figure 14.* **Effect of radius and $|\mathcal{D}_c|$ at fixed $K = 0.60$.** cACC (top), effective circuit size (middle), and abstention rate (bottom) as a function of certified radius for $|\mathcal{D}_c| \in \{50, 100, 500\}$. Larger $|\mathcal{D}_c|$ stabilizes both cACC and abstention at high radii; effective circuit size stays roughly constant. ResNet-101, $\tau = 0.95$, $n = 1,000$.

**Effect on circuit size and abstention at fixed $K$.** Fig. 14 fixes the circuit size at $K = 0.60$ and examines how cACC, effective circuit size, and abstention rate vary with radius for three concept dataset sizes. Three patterns are visible. First, at small $|\mathcal{D}_c| = 50$, cACC drops sharply as $r$ grows past $\sim 3$, while abstention rises toward 50–75% — the certificate becomes too aggressive for the available data. Second, at $|\mathcal{D}_c| = 100$ the cACC remains close to the baseline up to $r = 5$ and then drops. Third, at $|\mathcal{D}_c| = 500$ cACC is stable across the full radius range $r \in [1, 11]$, with abstention staying below 30%. Effective circuit size (middle row) remains roughly constant in all three settings, indicating that the certification budget is

spent on abstaining from unstable components rather than shrinking the certified set.

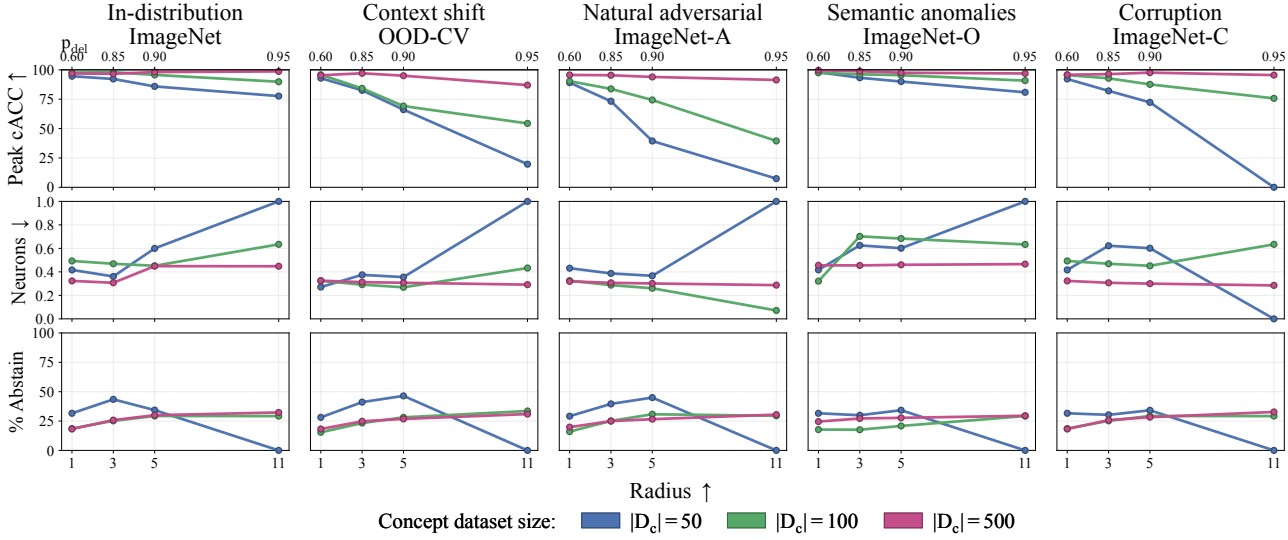

*Figure 15.* **Effect of radius and $|\mathcal{D}_c|$ at the per-setting optimal $K$.** Peak cACC (top), corresponding circuit size (middle), and abstention rate (bottom) as a function of certified radius. With $K$ optimized per setting, higher radii yield more compact circuits without sufficiency loss when $|\mathcal{D}_c|$ is large enough. ResNet-101, $\tau = 0.95$, $n = 1,000$.

**Effect at the per-setting optimal $K$.** Fig. 15 repeats the analysis with $K$ optimized per $(r, |\mathcal{D}_c|, \text{dataset})$ setting, reporting peak cACC. The picture is consistent with the fixed-$K$ analysis but more favorable: with the freedom to shrink the circuit at higher radii, peak cACC degrades less, and the optimal circuit size visibly contracts as $r$ grows (middle row). This is the behavior we want from a certification procedure: at higher radii, more components are flagged as unstable and abstained from, yielding more compact circuits with no sufficiency penalty. At $|\mathcal{D}_c| = 500$, peak cACC is essentially flat across $r \in [1, 11]$ on every benchmark.

**Takeaway.** Certified vision circuits scale to large radii given sufficient concept data: with large enough $|\mathcal{D}_c|$, certified circuits outperform baselines on both ID and OOD at $r$ up to 59, while providing stronger guarantees.

### F.5. Comparison to Vanilla Majority Vote

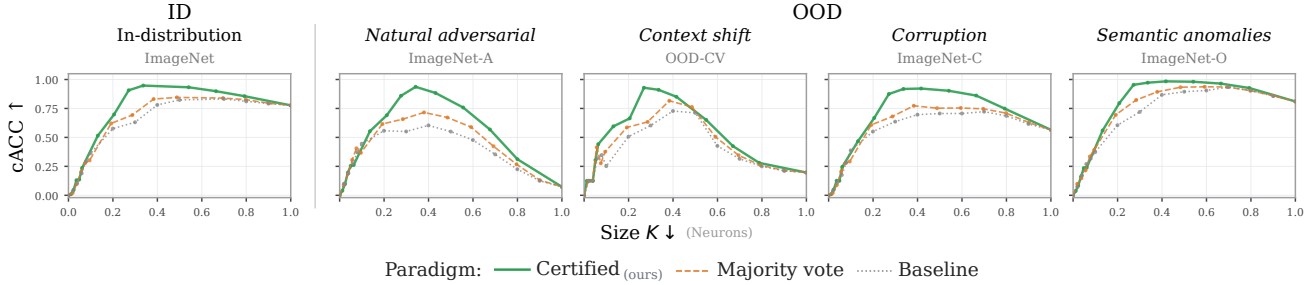

*Figure 16.* **Certified circuits outperform majority vote at identical compute.** cACC vs. $K$ for certified, majority vote (same $n$, >50% inclusion, no $\tau$), and baseline across five benchmarks. The gap between certified and majority vote isolates the contribution of certification (through the threshold $\tau$) beyond ensemble averaging.

A natural question is whether the gains of certified circuits come from certification itself, or simply from the ensemble averaging effect of running the base algorithm on $n$ subsamples. To isolate this, we compare certified circuits against a majority vote baseline at matched compute. Majority vote uses the same $n = 1,000$ deletion samples but includes each neuron appearing in more than $50\%$ of sampled circuits, with no confidence threshold — isolating the contribution of $\tau$ from ensemble averaging alone. Fig. 16 shows certified circuits outperform majority vote by a wide margin across all five

benchmarks in ID and OOD, with the largest gaps on OOD-CV and ImageNet-A where instability is most pronounced. This is notable because prior certification methods typically trade accuracy for robustness (Lécuyer et al., 2019; Cohen et al., 2019; Fischer et al., 2021; Anani et al., 2024), whereas certified circuits improve both. Majority vote already improves over the baseline thanks to averaging, but stops short of the certified circuit, indicating that $\tau$ contributes beyond ensemble averaging by excluding neurons whose inclusion votes are split rather than majority-aligned. These are the neurons most likely to be spurious, and abstaining from them is what makes the certified circuit both more accurate and more compact. We provide further feature visualizations of such neurons in the next App. F.6.

## F.6. Additional feature visualizations

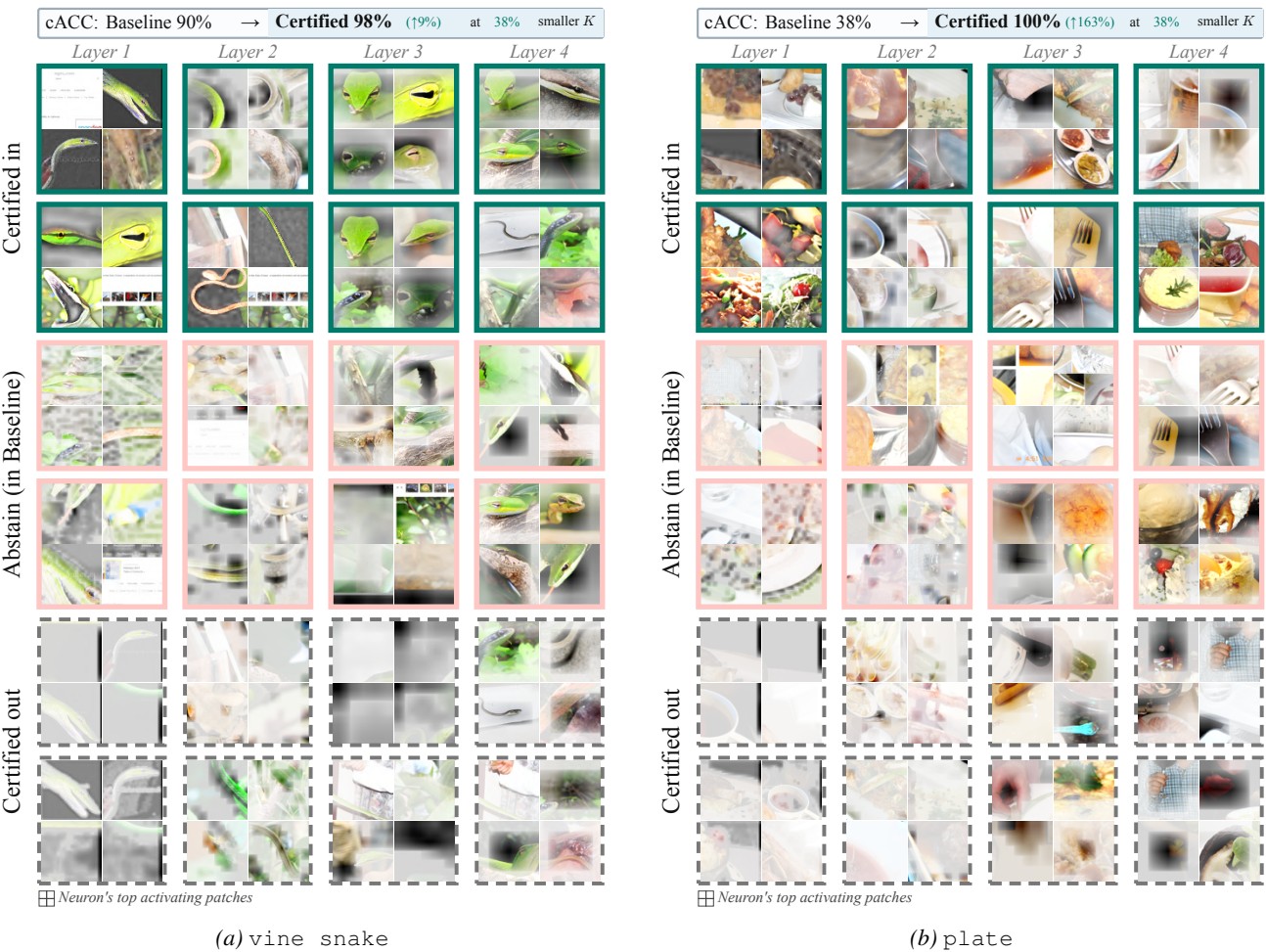

*Figure 17.* **Features encoded by certified-in, abstain (within baseline), and certified-out neurons** (ResNet-101). Per layer, top-activating neurons in each category. Certification improves cACC while shrinking the circuit by abstaining from unstable, spurious neurons.

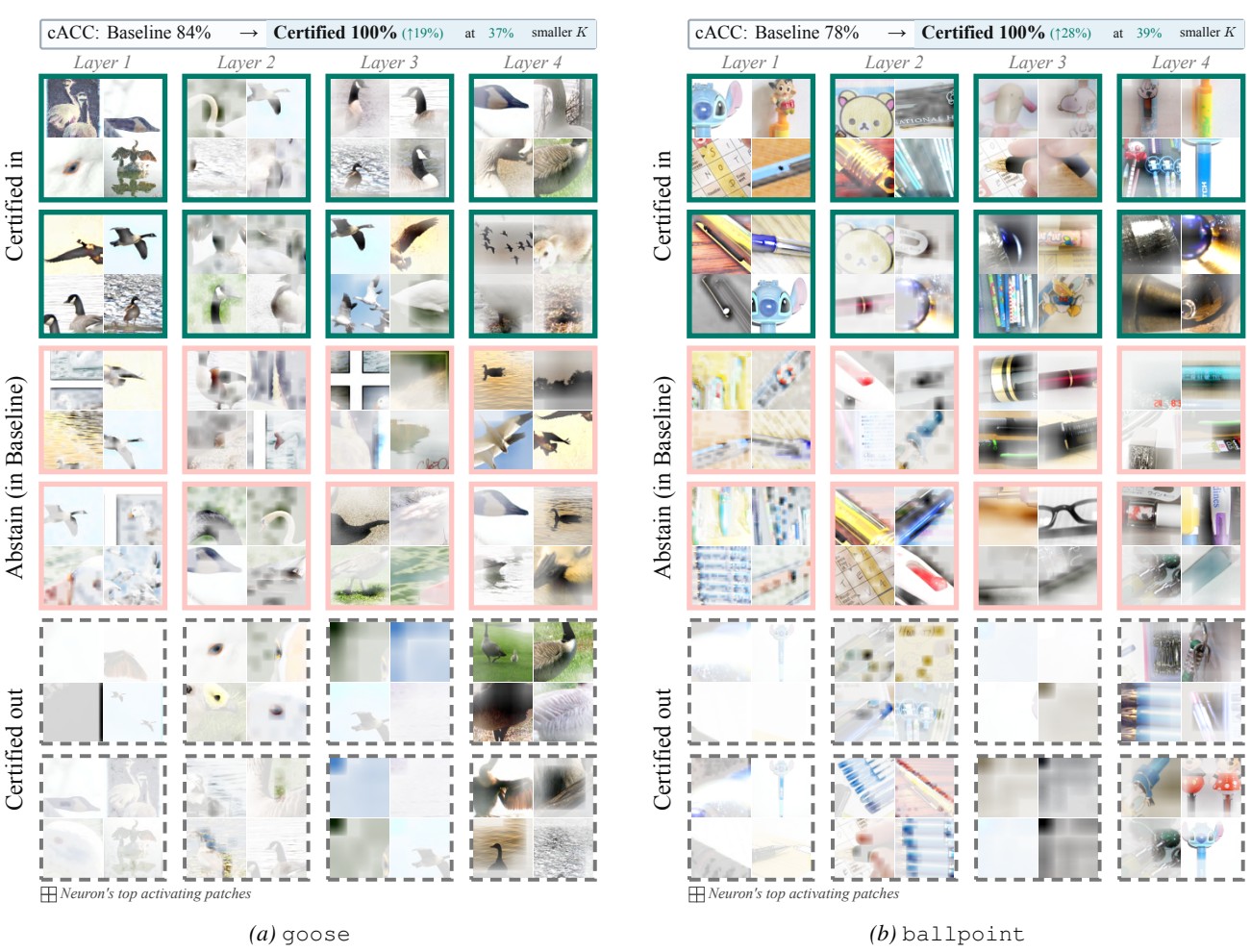

*(a)* `goose`                                    *(b)* `ballpoint`

*Figure 18.* **Features encoded by certified-in, abstain (within baseline), and certified-out neurons** Extension of App. Fig. 17

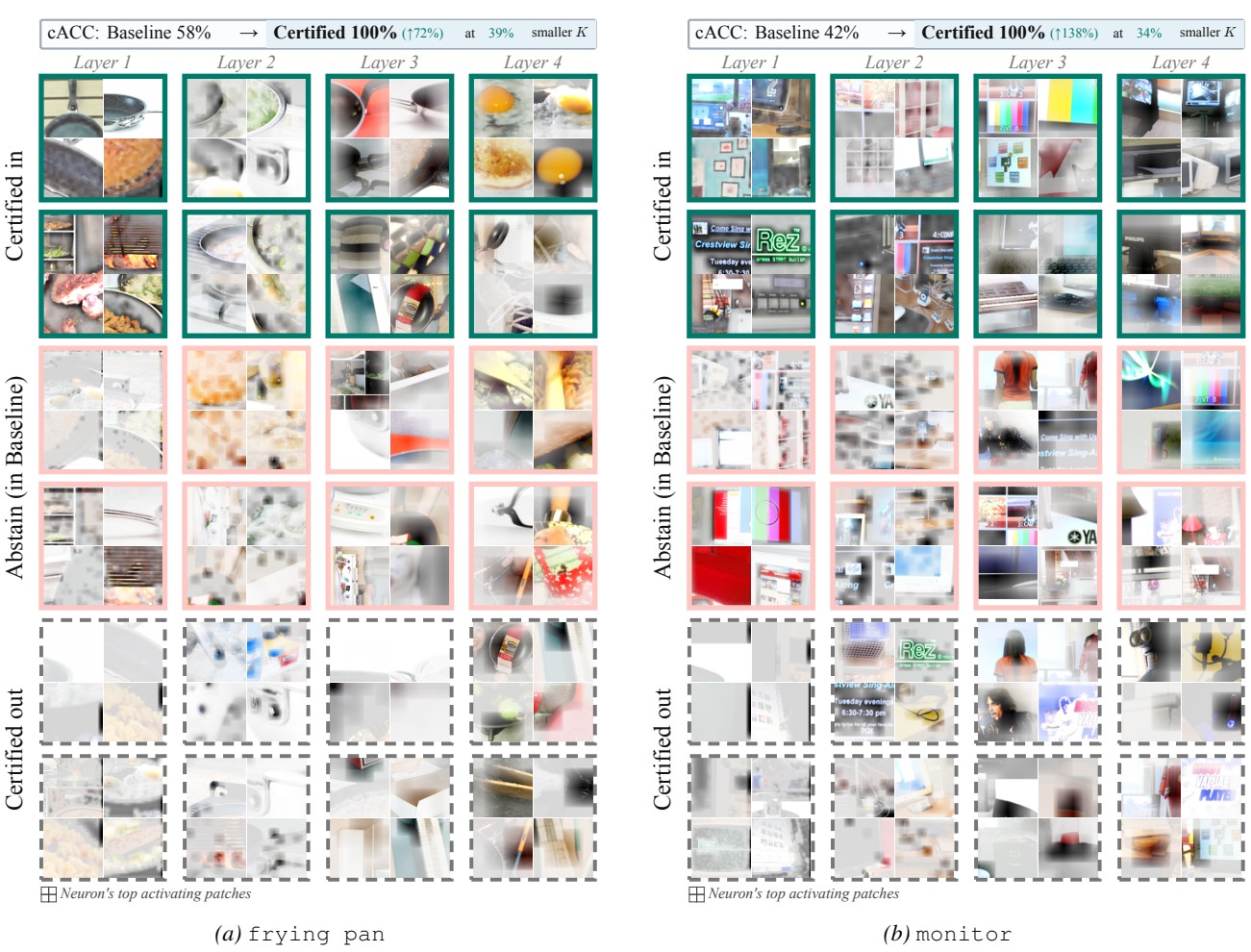

*(a)* `frying pan`    *(b)* `monitor`

*Figure 19.* **Features encoded by certified-in, abstain (within baseline), and certified-out neurons** Extension of App. Fig. 18

