# OpenReview forum: "Certified Circuits: Stability Guarantees for Mechanistic Circuits"
_ICML.cc/2026/Conference — ICML 2026 regular_

### Official Review · Reviewer_oB6d · 2026-03-11

**Soundness:** 4
**Presentation:** 4
**Significance:** 3
**Originality:** 2
**Overall Recommendation:** 5
**Confidence:** 3

**Summary:**

The paper proposes a technique to improve the robustness (quality) of circuit discovery algorithms for mechanistic interpretability and presents experimental results.
The method, very similar to randomized smoothing, uses a given base dataset and circuit discovery algorithm and runs the algorithm on many sampled subsets.
Then the procedure applies a majority vote for each vertex in the identified circuit, abstaining where the base algorithm gives inconsistent classification.
The experiments show that the resulting circuits are indeed more stable in terms of edit distance and more robust, e.g., in OOD settings.

**Compliance With Llm Reviewing Policy:**

Affirmed.

**Final Justification:**

The rebuttal addressed the two concerns about technical details: runtime and statistical aspects.
The authors clarified that runtime overhead is not necessarily naive and can be substantially reduced by extensive reuse of compute via caching.
The authors also clarified how statistical significance is controlled in their method and stated that they will emphasize this in the manuscript.

With these two points clarified, I think this manuscript is solid in motivation and execution, and the limitations of the approach are now clear.

**Key Questions For Authors:**

**Q1: Influence of $\alpha,n$ on algorithm output**
Could you spell out how tight/loose the lower bounds are as a function of $\alpha$ and $n$? e.g., the specific bounds used for experimental results.
How does this influence the circuit size? For which choices of $\alpha,n$ will these bounds be vacuous?

**Q2: Union bounds over circuit sizes**
How important would you judge controlling the statistical significance of the "full circuit"?
What are the typically found circuit sizes, and with a union bound, how large can $\alpha$ be for reasonable $n$ to result in tractable runtime?

**Q3: Runtime improvements with engineering**
Related to one of my statements in **W2:** Do you think the runtime cost can be significantly reduced with low-effort engineering methods such as caching?
I understand this might be specific to the underlying algorithm, but it would potentially result in much better runtime.

**Q4: Actual Runtime Overhead in Experiments**
Do I understand correctly that the runtime of your method is $n=1000$ times that of the baselines?
How long does a typical run for these baselines take, approximately at least?

**Limitations:**

Yes.

**Strengths And Weaknesses:**

The premise of this paper is very simple, to the point of seeming trivial.
However, the manuscript is **very polished, well written, and executed**, and it would be unjust to fault it for its underlying simplicity.

The only criticism I have is that the manuscript **does not emphasise its runtime cost and statistical significance sufficiently**.
I am not familiar with current work in circuit discovery and therefore feel **not confident ** to judge the impact of the manuscript.**

**In Summary, I score this manuscript as a weak accept, with strong execution of a simple idea, but drawbacks that are not clearly emphasized.**

## Strengths

**S1 Well Written and Executed**: As mentioned above, the manuscript is very polished and clear in writing. It is easy to read; the problem, motivation, and significance are well-motivated.
This (mostly) extends to the proof in the appendix as well.

**S2 Convincing Experimental Results:** The experiments illustrate that the method increases the quality of the results.
The observed effects are not surprising to me, but the experiments illustrate this effectively.
## Weaknesses/Limitations

**W1 Missing Emphasis of Statistical Parameter $\alpha$:**

The method (seemingly) relies on confidence intervals with significance $\alpha$.
This $\alpha$ controls---if I understand correctly---how "easily" the model abstains, depending on the number of samples $n$.
This impact and the precise numerical relation seem significant, yet the manuscript does not explain what $\alpha$ does in Theorem 3.1, and the meaning is not spelled out clearly.

In this regard, the last line in the appendix reads almost alarmingly: the statistical confidence is controlled for each vertex in the circuit, and a union bound would be needed to control it over the full circuit.
As a reader, this statement seems a bit like an attempt to hide just how many samples $n$, would be needed to yield a meaningful (statistical) guarantee.


**W2 Missing Emphasis and Exploration of runtime overhead/sample size $n$**

This is strongly related to **W1**, but I believe the runtime needs additional emphasis to frame the experimental results fairly:
**Does a, e.g., $n$=1000-fold increase in runtime justify the improved quality of the solution?**
I think it actually hurts the manuscript that this cost is not weighed against the benefit of the results:
Even though it is a black-box procedure, I would assume that, with little engineering effort, partial results from individual circuit discovery runs could be reused (e.g., caching activations between runs?) and drastically reduce runtime overhead.

**Small notes:**
- In Theorem 3.1. I believe the statement, as written, is not quite what was intended.
This statement cannot be a formal equality, but rather an implication, or an equality to the non-abstaining algorithm like this: $\tilde A_v^{\tau}(D)=\tilde A_v(D')$.
(otherwise this would imply $\tilde A_v^{\tau}(D)$ to be constant for all $D$ when $r$ is nontrivial)

- Theorem 3.1. does not explain what alpha is and how it scales with the number of samples/discovery runs

- This is a personal opinion, but the manuscript oversells the idea a bit.
I would tone down the introduction, conclusion, and impact statement slightly; it reads hyperbolically.

---

> ### Author Rebuttal · Authors · 2026-03-31
>
> Dear reviewer oB6d,
>
> We thank you for the thoughtful review and are glad you found the manuscript **well-written and executed**, the **experimental results convincing**, and the **soundness excellent**. We address each weakness and question below, with new figures provided as PDF https://anonymous.4open.science/r/icml26-certified-circuits/rebuttal_new_figures.pdf.
>
> **Role of $\alpha$ and union bounds (W1, Q1, Q2)**
>
> To clarify, $\alpha$ bounds the **familywise Type I error rate (FWER)** — the probability of *at least one* false certification (i.e., incorrectly certifying a circuit neuron/edge as meeting the probability threshold $\tau$). After Bonferroni correction over $N$ candidate components (the set of edges/neurons, depending on the circuit discovery method and architecture), each component is tested at significance level $\alpha/N$, ensuring the overall FWER is bounded by $\alpha$. For example, $N=3,840$ for ResNet-101 with Top-K neuron discovery. In response to your question, we now provide additional Figure 12, which shows the minimum number of samples $n$ required as a function of $\tau$ for $\alpha \in \{0.01, 0.001, 0.0001\}$, with and without the union bound.
> Key findings: (1) at $\tau=0.9$, the union bound costs only $\approx 2\times$ additional samples ($n=154$ vs $76$ for $\alpha=0.001$); (2) at $n=1000$, certification succeeds for $\tau^* > 0.985$ even at $\alpha=0.0001$. We also show in Figure 11 that varying $\alpha$ across $\\{0.01, 0.001, 0.0001\\}$ has minimal effect on the certified circuit cACC, confirming the bounds are not vacuous in practice at smaller $\alpha$. We will clarify this role of $\alpha$ in Theorem 3.1 in the revision and include the additional experiments, as well as the effect of increasing the number of components $N$ on the multiple hypotheses tests and union.
>
> **Runtime (W2, Q3, Q4)**
>
> Sorry for the misunderstanding, the runtime is *not* $n\times$ the baseline. Our implementation vectorizes the subsampling: we cache neuron scores once per image, then apply all $n$ deletion masks via batched matrix multiplication. This reduces sample-circuit time from a naive $\sim$1007s ($n=1000$) to $\sim$2.2s — a $\times$416 speedup (new Table 2). Total cost of discovering certified circuits is $\sim2.4$s per class vs $1$s for the baseline — a $\sim 2.4\times$ overhead, not 1000$\times$. The certification decision rule adds only 0.19s. We now include new Table 2 with a full breakdown. We also note that $n$ can be as low as $\sim$100 with minimal degradation (new Figure 13 shows cACC$=$99 at $n=100$, identical to $n=1000$). We thank you very much for raising this point since it strengthens our narrative and we promise to include it with more detail in the manuscript.
>
> **Theorem 3.1 notation (small note)**
>
> Thank you — we agree the statement should be an implication rather than equality when the algorithm abstains. We will correct this in the revision.
>
> **Tone (small note)**
>
> We appreciate the feedback and will tone down the introduction and conclusion accordingly.
>
> **New experiments:** (1) $\alpha$ effect on certified circuit sufficiency analysis (new Fig. 11), (2) Minimum number of samples under union bound analysis with Bonferroni over $N=3{,}840$ against $\tau$ (new Fig. 12), (3) runtime breakdown with $\times$416 speedup (new Table 2), (4) effect of $n$ on circuit accuracy and abstention rates (new Fig. 13).
>
> **Additional experiments (prompted by all reviews) that may also be of interest:**
>
> We also provide: (1) **larger radii**: certified circuits outperform baselines even at $r=59$ with $|D_c|=1000$ (Figs. 4–6); LLM tasks reach $r=59$; (2) **EAP-IG on GPT2-Small** for 3 tasks, with up to $80\%$ fewer edges and +6pp accuracy gain (new Figs. 1–3); (3) **ViT-B/16** results (new Fig. 9); (4) **majority vote baseline** comparison confirming certification outperforms ensembling alone (new Fig. 10); (5) **uncertainty estimates** with significance tests ($p<0.001$, new Table 1); (6) **Certified circuits structural stability across seeds** (IoU$=0.973$, new Fig. 8); (7) **feature visualizations** of certified and abstain neurons (new Fig. 7); (8) **necessity analysis** (new Fig. 14). Results now span 3 architectures (CNN, ViT, LLM) and 2 discovery algorithms (Top-K, EAP-IG).

---

> > ### Author Rebuttal · Reviewer_oB6d · 2026-04-04
> >
> > I thank the authors for their clarifications; my concerns are partially addressed.
> >
> > First, regarding runtime: I think this aspect deserves more emphasis. I did not see a mention of this in the manuscript, and the beginning of the limitation section should actually be more optimistic in this case.
> > Something like "it scales linearly in principle, but we can reuse a lot of computation for most underlying algorithms, so runtime overhead is *actually* not a lot."
> > To me, this makes the approach much more usable at a glance. If I just need to wait twice as long for a circuit discovery algorithm to finish, I might as well run the "certified" version.
> >
> > For the statistical significance, I am not sure if I understand the following correctly:
> > - In Fig.12, does the $\alpha$ (with the union bound) mean $\alpha/N$ for $N=3840$?
> > - The information in Fig. 12 seems slightly misleading to me, as it is unclear for which $\tau: r<1$ (the guarantee is vacuous). I understand $p_{del}$ can be changed to avoid this, but still. If I understand correctly, for $p_{del}=0.6$ as in the experiments, Figure 12 is vacuous for $\tau>.9$, right?
> > - Does Fig 12 plot the all success Clopper-Pearson confidence interval, i.e., $n\geq \frac{\log \alpha}{\log \tau*}$? This seems to be consistent with the reported numbers. (and the N=3840 correction).
> > Do the assumptions of that confidence bound really map to your setting? Maybe I am confused here, but I thought this only works when you observe $\hat p=1$, and not $\hat p>tau$.
> >
> > In general, could you just write down here on OpenReview the used function $r > f(\alpha,\tau,p_{del})$ and $n > g(\alpha,\tau)$? I don't want to be pedantic, but this is slightly opaque, and I want to be sure the right statistical bound is used.
> >
> > In general, the statistical aspect is still unclear to me. I would appreciate it if the authors could clarify or provide a reference.
> > This is the only concern I have left.

---

> > > ### Author Response · Authors · 2026-04-04
> > >
> > > Thank you for carefully reading the rebuttal. We are glad to address your remaining concern.
> > >
> > > First, regarding runtime, we agree that this point deserves more emphasis. We will make this clearer in the paper and include the additional runtime analysis. Thank you so much for prompting this analysis, as it strengthens the paper by showing that the framework is efficient relative to the baseline.
> > >
> > > Regarding the statistical significance:
> > >
> > > - Yes, in Fig. 12 the solid lines use the Bonferroni-corrected level $\alpha / N$ with $N = 3840$. The dashed line for $\alpha = 0.001$ corresponds to the single-component case $N=1$, i.e., without the Bonferroni correction. We include this dashed line only as a reference; our actual statistical test uses the Bonferroni correction $\alpha/N$ because the guarantee must hold simultaneously over all components while controlling the total failure probability $\alpha$.
> > > - The main message of the new Fig. 12 is to show the minimum number of samples $n$ needed to certify at different values of $\tau$. As $\tau$ increases, certification becomes stricter, and therefore more samples are required. To address your specific question: for $p_{\text{del}} = 0.6$, Appendix Fig. 5 shows that the certified radius is $r=0$ for $\tau \leq 0.85$. Numerically, from the radius equation, one needs roughly $\tau \geq 0.9$ before the radius becomes nonzero, which is why the appendix figure shows $r=0$ for smaller values of $\tau$.
> > > - Yes, exactly! Fig. 12 plots the success confidence interval in the perfect-agreement case, i.e., when every neuron attains the maximum possible number of votes. This corresponds to the setting where the relevant smoothed probability passes the threshold in Eq. 4. Concretely, the smoothed algorithm assigns a non-abstaining label when either $p_v(\mathcal{D}) > \tau$ or $1 - p_v(\mathcal{D}) > \tau$.
> > >
> > > Regarding your final question, the certified radius $r = f(\tau, p_{\text{del}})$ is given directly by Theorem 3.1 (Eq. 5):
> > >
> > > $r = \left\lfloor \frac{\log(1.5 - \tau)}{\log p_{\text{del}}} \right\rfloor$
> > >
> > > Thus, $r$ depends on $\tau$ and $p_{\text{del}}$, and not on $\alpha$. At fixed $n$, $\alpha$ determines whether certification of a chosen radius is statistically feasible, but it does not alter the radius formula itself. Rather, $\alpha$ enters only through the statistical estimation step.
> > >
> > > For the minimum number of samples $n = g(\alpha, \tau)$, Fig. 12 considers the perfect-agreement case and plots the minimum total sample count required after Bonferroni correction over $|\mathcal{V}|$ vertices. In idealized form, this scales as
> > >
> > > $n > \frac{\log(\alpha / |\mathcal{V}|)}{\log(\tau)}$.
> > >
> > > In the actual plotted implementation, we use the corresponding integer threshold together with the additional offset $n_0$, i.e.,
> > >
> > > $n_{\mathrm{total}} = \left\lfloor \frac{\log(\alpha / |\mathcal{V}|)}{\log(\tau)} \right\rfloor + 1 + n_0$,
> > >
> > > where $n_0 = 10$ in Fig. 12 is the initial set of Monte Carlo samples used to estimate the top class before the certification step, following Fischer et al.
> > >
> > > Regarding the reference: We follow exactly the same multiple-testing correction as Fischer et al., Scalable Certified Segmentation via Randomized Smoothing (which we reference in the paper as well throughout Section 3), since our setting is directly analogous when circuit discovery is cast as a binary segmentation problem over components.
> > >
> > > We will incorporate this analysis into the paper and further clarify the roles of $\tau$, $p_{\text{del}}$, $\alpha$, and the minimum sample size $n$, including through the new Fig. 12 and the discussion above. We will also add pseudo-code for our certified circuits algorithm to make the procedure easier to follow, and we will share the codebase to support clarity and reproducibility.
> > >
> > > We are happy to discuss further if you have more questions.
> > >
> > > Thank you again for your time and for the sharp questions.

---

### Official Review · Reviewer_y47T · 2026-03-13

**Soundness:** 2
**Presentation:** 3
**Significance:** 3
**Originality:** 2
**Overall Recommendation:** 4
**Confidence:** 2

**Summary:**

The paper studies the problem of dataset overfitting in neural network circuit discovery, where extracted circuits often fail to generalize across datasets. To address this, the authors adapt randomized smoothing to the discrete dataset setting to obtain certified circuits. The approach repeatedly runs a circuit discovery method on perturbed datasets within a given radius and aggregates the resulting circuits using majority voting over nodes. This produces circuits that are provably stable to small dataset perturbations. Experiments on ImageNet distribution-shift benchmarks show improved out-of-domain performance and sparser circuits compared to standard discovery methods.

**Compliance With Llm Reviewing Policy:**

Affirmed.

**Key Questions For Authors:**

See the weaknesses section

**Limitations:**

Yes, the authors address the limitation.

**Strengths And Weaknesses:**

**Strengths**
- Circuit discovery is a key technique in mechanistic interpretability, and existing methods often overfit to the dataset used during discovery. Addressing robustness and generalization of discovered circuits is important problem to analyse.
- The empirical study includes several circuit scoring rules and evaluates performance under distribution shift using datasets such as ImageNet-A, ImageNet-O, and ImageNet-C. The proposed method consistently improves OOD accuracy and produces sparser circuits compared to the baseline approaches.

**Weaknesses**
- The certification result only considers a perturbation radius of $r=1$, corresponding to removing a single example from the dataset. This is an extremely small perturbation and it is unclear how such a guarantee relates to the large distribution shifts considered in the experiments.
- The certification procedure requires many Monte Carlo runs of the base circuit discovery algorithm. The paper mentions that these runs can be parallelized but does not provide concrete runtime or resource measurements, making it difficult to assess the practicality of the approach.
- The method repeatedly runs the base circuit discovery algorithm and aggregates the outputs. Since the dataset perturbations are minimal, the improvements may largely arise from repeated runs and aggregation similar to bagging or ensembling, rather than from the certification mechanism itself. A  baseline method with a similar computational budget without certification would help better understand the true souce of improvement.

---

> ### Author Rebuttal · Authors · 2026-03-31
>
> Dear reviewer y47T,
>
> Thank you for the constructive review and for recognising the **consistent OOD accuracy improvements** and **sparser circuits across multiple scoring rules and benchmarks**. We address each weakness below, with new figures provided as PDF https://anonymous.4open.science/r/icml26-certified-circuits/rebuttal_new_figures.pdf.
>
> **Certified radius and connection to OOD shifts (W1)**
>
> Even though we only report at $r=1$ in the manuscript, the certified radius is not fixed at $r=1$; it scales with $p_\mathrm{del}$ and $\tau$. We now provide a comprehensive sweep across $p_\text{del}\in\\{0.6, 0.85, 0.95, 0.99\\}$ corresponding to  $r\in\\{1, 3, 11, 59\\}$, each for $|D_c| \in \\{50, 100, 500, 1000\\}$, reporting cACC, circuit size, and %Abstain across all 5 benchmarks (new Figures 4–6). Even at $r=59$ with $|D_c|=1000$, certified circuits exceed baseline cACC on every benchmark. The connection to OOD generalisation is direct: certified-in neurons are those whose inclusion is stable under dataset perturbation. These are precisely the concept-relevant neurons that transfer across distribution shifts, while abstained neurons are the ones that overfit to the specific discovery images (we qualitatively visualize highlighted features by both category of neurons in new Fig. 7 and Figs. 16-17). Our new LLM experiments (new Figures 1–3) further illustrate this: GPT-2-Small tasks naturally yield certified radii up to $r=59$ due to larger concept datasets, with consistent accuracy and compactness gains.
>
> **Runtime and practicality (W2)**
>
> We now include a full runtime breakdown in new Tab. 2. Through vectorized activation/relevance (and any neuron score) caching, our optimized implementation reduces sample-circuit time from a naively expected  $1007$s ($n=1000$) to $\sim 2.226$s, a $\times 416$ speedup. The certification decision rule adds only $0.194$s per circuit. Total cost: **$\sim 2.42$s per class** for $n=1000$ on ResNet-101, compared to $1.007$s for the single-circuit baseline, a $\sim 2.4\times$ overhead that yields substantially more compact and OOD-robust circuits. We will include this runtime discussion in more details in the manuscript.
>
> **Ensemble effect vs. certification (W3)**
>
> We directly compare certified circuits against a majority-vote baseline using the same n subsamples and >50% inclusion rule, without the certification threshold $\tau$ (new Fig. 10). Certified circuits consistently outperform majority vote on both ID and OOD benchmarks at identical compute. This isolates the contribution of $\tau$: by requiring statistically confident inclusion votes rather than a bare majority, the certified method more aggressively removes neurons whose membership is ambiguous, and these are disproportionately the spurious, non-transferable ones. We further qualitatively visualize the certified-in neurons and the abstain ones that are included by the baseline in new Figure 7 and Figures 16-17.
>
> **Generality and scope of contribution**
>
> Beyond the analyses above, we have added new experiments on EAP-IG (edge attribution patching) applied to GPT-2-Small (3 tasks: IOI, IOI Hard, Greater-Than) (new Figures 1-3) and ViT-B/16 (new Figure 9). The framework requires minimal adaptation, certifying edges instead of neurons, with consistent improvements (up to $80\%$ fewer edges, +6pp OOD accuracy). Results now span **3 architecture families (CNN, ViT, LLM) and 2 discovery algorithms (top-K, EAP-IG)**, confirming that the contribution extends beyond a narrow adaptation of randomized smoothing.
>
> **Additional analyses (prompted by other reviews)**
>
> We provide uncertainty estimates (mean $\pm$ SE across 100 classes) with significance tests in new Tab. 1; most improvements are significant at $p < 0.001$. We also verify certified circuit structural stability: pairwise IoU across 40 random seeds yields a mean of $0.973$ even for the least stable classes (new Fig. 8).
>
> In summary, new experiments now span 3 architectures, 2 types of algorithms, comprehensive ablations, and radius/abstention analysis, addressing the raised concerns and further strengthening the contribution. We will integrate all additions into the revised manuscript.

---

> > ### Author Rebuttal · Reviewer_y47T · 2026-04-04
> >
> > My questions have been adequately addressed, and I will retain my rating.

---

> > > ### Author Response · Authors · 2026-04-04
> > >
> > > Thank you for going through the rebuttal and acknowledging that the rebuttal adequately addressed all your questions. We kindly ask whether you might consider raising the score to reflect the resolved concerns. Should any concerns remain, we would appreciate if you point them out so that we have a chance to improve the paper further.
> > >
> > > Thank you again for the constructive review and careful reading of the paper.

---

### Official Review · Reviewer_sQ6c · 2026-03-13

**Soundness:** 4
**Presentation:** 4
**Significance:** 3
**Originality:** 3
**Overall Recommendation:** 5
**Confidence:** 4

**Summary:**

Circuit discovery is a subfield of mechanistic interpretability, whose goal is to recover sparse subgraphs of the network (circuits) that explain a specific behavior. However, recent evidence has documented that those circuits are unstable. In particular, small changes to the concept dataset may produce completely different circuits. The authors address this issue by proposing Certified Circuits, a framework that can extend any existing circuit discovery technique by adding randomized data subsampling to it, using RS-Del and per-component certification from randomized smoothing. Their method aggregates votes across random sub-datasets for each neuron and certifies it as in (accepted), out (rejected) or abstain (no robust decision can be made), giving a provable guarantee that neuron membership is invariant under bounded edit distance perturbations of the initial concept dataset. The authors show experimentally on ResNet and across multiple benchmarks (ImageNet + 4 OOD datasets) that certified circuits are more compact (up to 45% smaller), more accurate (up to 91% higher cACC), and more stable (tested on shifted distributions) than their uncertified equivalents.

**Compliance With Llm Reviewing Policy:**

Affirmed.

**Final Justification:**

My initial reason for a 4 was almost entirely due to soundness, which required additional details/experiments.
The rebuttal has fully addressed my concerns, and I have increased my score from 4 to 5 and confidence from 3 to 4.
See the rebuttal acknowledgement for details.

**Key Questions For Authors:**

1. Can you provide confidence intervals or standard errors for cACC in Table 1 across the classes, as well as IoU between circuits from independent runs of the certification procedure with different random seeds? This would directly address whether the gains are statistically reliable (crucial for a paper on robustness and stability) and whether certification converges to a stable solution.
2. How do cACC/circuit size/abstention rate change as you increase p_del and \tau to get r > 1? Does the framework scale to practical meaningful values of the radius, or is it limited to r=1?
3. As mentioned above, can you provide any evidence for non-top-K discovery, even on a small-scale experiment?
4. Can you provide a qualitative analysis or feature visualizations that would show that abstained neurons correspond to dataset-specific features rather than concept-relevant ones with borderline scores?

**Limitations:**

The authors discuss computational cost, CNN-only scope, the deletion-rate trade-off, and the single sparsity K across layers. A brief discussion of the very small certified radius and its practical implications would improve this section.

**Strengths And Weaknesses:**

The motivation of the paper is clear and justified: the authors address the problem of the instability of discovered circuits under changes to the concept dataset, which is part of the many robustness issues that have recently been uncovered in mech interp, and particularly circuit discovery. The authors' core idea is to wrap any black-box circuit discovery algorithm with randomized subsampling, certifying inclusion decisions for each neuron using RS-del and segmentation-style abstention, which is original and sound. To make the problem tractable, the authors formalize circuit discovery as a dataset to binary mask mapping, using edit distance as the main threat model, which is realistic. The mathematical derivations seem correct.

The experiments are well designed and structured around clear questions (sufficiency, compactness, OOD generalization and structural stability), and the authors' method produces consistent strong results across multiple datasets, scoring functions and architectures. I particularly appreciate the structural stability experiment (5.3) on measuring IoU between circuits discovered on ImageNet and rediscovered on OOD data, which is an excellent type of experiment that would not be included in many papers in the field. The authors show that certified circuits often outperform even the full unmodified model on OOD data, which is a strong argument for their proposed technique.

Another very interesting point of the paper is that the "abstain" decision for a given neuron is not simply "giving up" on producing an interpretation as one might expect. The authors show empirically that neurons that cannot be certified are in fact the ones that improve compactness and accuracy when removed. This is a strong find that would not be obvious from the theoretical framework alone.

In turn, the paper has several limitations:
- The framework is algorithm-agnostic. However, only top-K circuit selection is evaluated on ResNet. A lot of the cited circuit discovery literature focuses on transformers/LLMs (ACDC, activation patching, causal tracing). I understand that the focus of the paper is largely on image models, but it is unclear whether certification would work as well on more complex discovery methods (as top-K selection is rather crude). A, experiment with a different algorithm (e.g. edge pruning on a small transformer) would both validate the algorithm-agnostic claim more empirically and broaden the appeal of the paper.
- The authors choose p_del = 0.6 and \tau = 0.95, which produces a certified radius of r=1. Given the size of the concept datasets, a single edit seems like a very small perturbation. The experimental results do suggest that the circuits are stable for larger perturbations, but there is still a large gap between the theoretical guarantee and the observed results. It would be beneficial to show how accuracy/circuit size/abstention rate evolve when r increases, even if the performance is degraded, as this would be useful for practitioners to evaluate the practical utility of the method.
- The paper's contribution is fundamentally about robustness, but the data reported on Table 1 are point estimates averaged over 100 classes and do not contain CIs/standard errors/significance tests, which is regrettable. The paper could also measure the stability of the certification itself: if the pipeline is run multiple times for the n=1000 deletion mask with different seeds, how similar would the produced circuits be? It is worth noting that Méloux et al. show that circuit discovery can be non-identifiable (multiple different circuits can be found for the same behavior and dataset), while the paper here assumes that there exists a unique invariant core. However, the majority vote aggregation could potentially select degenerate solutions. Furthermore, if non-identifiability is inherent to circuit discovery, then it should manifest as high variance in the found circuits and should show up if an experiment such as the one mentioned above was conducted. I understand that this is a hard problem and that the authors seek practical solutions, but a brief discussion of the interaction between certification and non-identifiability would be useful.
- The authors claim that abstained neurons are spurious. This seems intuitive enough, and the accuracy improvements are consistent with it, but it is not directly verified. Since abstention is determined by vote consistency, a relevant neuron with borderline scores near the top-K threshold would also be abstained. Some qualitative analysis (e.g. additional detailed examples or probing) would make the mechanistic narrative stronger. Finally, the authors evaluate sufficiency but not necessity (also tied with non-identifiability: do the certified circuits capture the primary causal pathway, or one of potentially many sufficient subgraphs?)

Overall, this is a paper with significant contributions. It introduces a general (algorithm-agnostic) tool, provides the first formal dataset-level stability guarantees for circuit discovery, and shows that these guarantees are experimentally beneficial. The framework and experimental results are strong. The paper suffers from a few limitations listed above, but these can mostly be addressed through additional experiments. As such, I recommend weak acceptance, with potential for improvement if the limitations are addressed.

---

> ### Author Rebuttal · Authors · 2026-03-31
>
> Dear reviewer sQ6c,
>
> We thank you for the thorough review and are glad you found the **motivation clear**, **derivations correct**, and appreciated the **structural stability experiment** and the **finding that abstained neurons improve accuracy and compactness**. We address each point below, with new figures provided as PDF https://anonymous.4open.science/r/icml26-certified-circuits/rebuttal_new_figures.pdf.
>
> **Algorithm-agnosticism, Transformers, LLMs (W1, Q3)**
>
> We agree that circuit discovery also plays a major role in LLM literature and have now certified the EAP-IG algorithm (edge attribution patching with integrated gradients, based on ACDC) on GPT-2-Small, similar to the original paper, on 3 tasks: IOI, IOI (Hard), and Greater-Than (new Figs 1-3). Certified EAP-IG circuits match or improve peak accuracy at smaller sizes (up to $80\%$ fewer edges on Greater-Than OOD; $+6$ pp on IOI Hard OOD). Here, we treat attention edges rather than neurons as certified components. We also extend our architectures to ViT-B/16 (new Fig. 9), giving **strong certified circuit sufficiency and compactness across 3 architectures (CNN, ViT, LLM) and 2 discovery algorithms (top-K, EAP-IG)**.
>
> **Larger radii (W2, Q2)**
>
> We provide new analysis showing competitive performance of certified circuits at higher radii. We vary $p_{\text{del}}$ and $|D_c|$ across all 5 benchmarks (new Fig 4-6). The key finding is that *certified circuits outperform the baseline at higher radii, provided* $|D_c|$ *scales accordingly*: at $|D_c|=500$, cACC stays above $90\%$ even at $r=11$ (new Fig 4). Notably, at $|D_c|=1000$, *certified circuits outperform the baseline even at* $r=59$. For small concept datasets ($|D_c|=50$), accuracy drops to zero at larger radii ($r\in\\{11,59\\}$), since the high deletion probability effectively removes the entire dataset during subsampling. Circuit size decreases slightly with radius as more neurons are abstained (new Fig. 5, rows 2–3), yet accuracy remains high at $|D_c| = 1000$, confirming that abstained components are unnecessary for sufficiency. Crucially, *LLM EAP-IG certified circuits naturally achieve higher task accuracy at much larger certified radii* ($r$ up to $59-80$, new Fig 1, 3) due to larger concept datasets, demonstrating that the framework scales well beyond $r=1$.
>
> **Uncertainty estimates (W3, Q1a)**
>
> We extend Table 1 to report mean $\pm$ SE with paired Wilcoxon tests (new Table 1): all certified vs. baseline comparisons achieve $p<0.001$, with SEs small relative to gains (e.g., $96\pm1.0$ vs $86\pm1.2$).
>
> **Structural stability across seeds (W3, Q1b)**
>
> We compute pairwise IoU of certified circuits across 40 independent seeds (new Fig 8). The key finding is that *certified circuits exhibit high structural stability across seeds*, with mean IoU of $0.973 \pm 0.0015$ ($95\%$ CI) even for the 10 least stable classes.
>
> **Circuit non-identifiability (W4)**
>
> Thank you for raising this very interesting point. M'eloux et al. identify data resampling as a key contributor to circuit variance (cf. Bootstrap in their Tab. 1). Our certified majority vote operates over precisely such resampled subsets. It is designed to retain only those components that are *consistently* selected across data perturbations, filtering out the non-identifiable periphery. Certification directly addresses the instability that M'eloux et al. document: components that would differ across bootstraps are abstained rather than arbitrarily included. An additional seed stability analysis (new Fig. 8) confirms convergence to a stable core rather than degenerate solutions. Extending guarantees to e.g., prompts paraphrasing is future work; our EAP-IG experiments on GPT2-Small provide a proof of principle where certified circuits generalize to harder OOD prompts empirically.
>
> **Abstained neurons (W5, Q4)**
>
> We provide additional feature visualizations (new Fig 7), which show that certified neurons encode class-specific features (numbers for *digital clock*, body parts for *vine snake*), while abstained neurons respond to generic textures, confirming abstention targets spurious rather than concept-relevant features. We also provide randomly selected circuits in new Fig. 15-17.
>
> **Circuit necessity (W6)**
>
> We provide an additional progressive ablation of certified circuits (new Fig 14), which shows clear cACC drops, confirming certified-in neurons are genuinely important. The certified circuit drops slightly more, consistent with retaining a more concept-relevant core.
>
> **Small certified radius (L1)**
>
> See (W2, Q2) where r is increased. Practically, $r=59$ guarantees invariance to adding/removing/replacing up to 59 concept images, covering annotator disagreement and curation noise.
>
> ---
> We hope these additions address your concerns and would be happy to discuss further during the discussion period.

---

> > ### Author Rebuttal · Reviewer_sQ6c · 2026-04-02
> >
> > I thank the authors for this is comprehensive rebuttal and the substantial new experiments. All of my main concerns are adequately addressed:
> > - The fact that the method is algorithm-agnostic is more convincingly supported by the new experiments on EAP-IG on two architectures
> > - The analysis on larger radii demonstrates that the framework scales largely past r=1 when concept datasets are larger
> > - The added confidence intervals confirm that all gains are significant
> > - The seed-stability analysis addresses my concern about convergence.
> > - I appreciate the feature visualizations that quite convincingly show that certified neurons encode class features while abstained ones are more generic
> > - The necessity ablation confirms that the certified neurons are important.
> >
> > Due to the thorough response and new experiments, I am raising my score from 4 to 5 and my confidence from 3 to 4; this is a very solid paper.
> >
> > Minor note to authors: my review referenced Méloux et al. (2025) "Everything, Everywhere, All at Once: Is Mechanistic Interpretability Identifiable?", but since I did not include the title, it seems the authors referenced a different paper from the same author. The specific work I meant to cite shows that multiple different circuits can equally satisfy faithfulness metrics, which may be relevant to the authors' work. This is however not a requirement.

---

> > > ### Author Response · Authors · 2026-04-04
> > >
> > > Thank you for the thorough and constructive review, and for raising the score from 4→5 and confidence from 3→4 in light of our new experiments. We sincerely appreciate that you think this is a **very solid paper**. We are grateful for all the points raised by you which helped to strengthen the paper and have led to stronger and more complete findings about the certified circuits framework, and for also carefully reading the rebuttal.
> > >
> > > We thank you for clarifying the Méloux et al. (2025) reference on identifiability in circuit discovery. We in fact were referencing their other work on circuit variance "Mechanistic Interpretability as Statistical Estimation: A Variance Analysis". This is definitely an interesting direction and we are curious to test how certified circuits interact with non-identifiability, and whether there is a relation between abstentions and multiple sufficient circuits.

---

### Official Review · Reviewer_CHgM · 2026-03-13

**Soundness:** 3
**Presentation:** 4
**Significance:** 3
**Originality:** 3
**Overall Recommendation:** 4
**Confidence:** 3

**Summary:**

This paper introduces a framework that wraps any black-box circuit discovery algorithm with a randomised data subsampling procedure to produce provably stable mechanistic circuits. The core technical contribution is an adaptation of deletion-based randomized smoothing (RS-Del) to the circuit discovery setting: by repeatedly running a base discovery algorithm on randomly deleted subsets of a concept dataset and aggregating per-neuron inclusion votes, the framework certifies that each neuron's in/out decision is invariant to any concept dataset edit within a bounded edit-distance radius r. Neurons that cannot be certified are abstained from, yielding sparser circuits. The approach is algorithm-agnostic and model-agnostic. Experiments on ResNet-50 and ResNet-101 using ImageNet and four OOD benchmarks demonstrate that certified circuits are simultaneously more compact (up to 45% fewer neurons) and more accurate (up to 91% higher cACC on OOD data) than uncertified baselines, while also showing higher structural consistency when re-discovered on shifted distributions.

**Compliance With Llm Reviewing Policy:**

Affirmed.

**Key Questions For Authors:**

Q1. The experiments use pdel = 0.6 and τ = 0.95, yielding a certified radius of r = 1. This means the formal guarantee covers only datasets differing by a single edit. In practical circuit discovery, concept datasets often contain tens to hundreds of images, and practitioners might expect stability under much larger perturbations. Can the authors characterise for what dataset sizes and pdel settings r ≥ 5 or r ≥ 10 becomes achievable without substantially degrading base algorithm performance?

Q2. The paper claims algorithm-agnosticism but exclusively evaluates with top-K scoring as the base algorithm A. Does the framework provide comparable stability and accuracy improvements when wrapping more principled discovery methods such as ACDC or attribution patching?

Q3. Much of the OOD accuracy improvement may be attributable to the ensemble-style effect of running A on n=1000 subsamples and taking consensus, rather than to the formal certificate specifically. Have the authors compared against a simple majority-vote ensemble baseline that uses the same n subsamples but without the certification threshold τ?

Q4. The paper currently evaluates only on CNNs. Given that most active mechanistic interpretability work targets transformer language models, can the authors provide at least a preliminary characterisation of what challenges or adaptations would be required to apply Certified Circuits to, for example, GPT-2 attention head circuits?

**Limitations:**

yes

**Strengths And Weaknesses:**

Soundness

The theoretical foundation is well-grounded. Theorem 3.1 is a correct and appropriately scoped application of RS-Del (Huang et al., 2023) combined with the component-wise abstention approach from Fischer et al. (2021). The proof in Appendix A is complete and follows standard arguments from the randomized smoothing literature. The certified radius formula is correctly derived and the dependence on pdel and τ is clearly explained.

There are some soundness limitations worth noting. The certified radius r = 1 chosen in the experiments (with pdel = 0.6, τ = 0.95) is quite conservative hence certification is only guaranteed for concept datasets differing by a single edit. While the authors argue this is meaningful in practice, it significantly limits the scope of the formal guarantee, particularly for larger or noisier concept datasets where practitioners might expect a wider stability margin. The paper does not adequately characterize what fraction of neurons are abstained from under these settings, nor does it investigate whether the abstain rate changes substantially with dataset size. Additionally, the evaluation uses a relatively simple top-K scoring function as the base circuit discovery algorithm A; it is unclear how well the results transfer to more sophisticated discovery methods like ACDC or attribution patching, despite the algorithm-agnostic claim.

Presentation

The paper is well-written and logically structured. The problem motivation (circuit instability due to dataset sensitivity) is compelling and well-supported with citations. Figure 2 clearly communicates the multi-step certification pipeline. The theoretical framework is presented concisely but with sufficient formalism. The experimental setup, metrics (cACC, oACC, IoU), and baselines are clearly defined. The impact statement explicitly notes the risk of certified circuits being misinterpreted as complete accounts of model behaviour.

Significance

Circuit instability is a genuine and well-documented problem in mechanistic interpretability, and providing formal guarantees for circuit discovery is a meaningful contribution. The OOD generalisation results are impressive, achieving 94% cACC on ImageNet-A (natural adversarial examples) compared to 62% for the baseline is a striking improvement, and the simultaneous gain in compactness strengthens the narrative that abstaining from unstable neurons removes spurious rather than concept-relevant features. The algorithm-agnostic, model-agnostic framing is practically appealing.

However, significance is tempered by several factors. The experiments are confined to CNNs (ResNet-50/101); the paper acknowledges that extension to ViTs and LLMs is future work, which is a significant gap given that most active mechanistic interpretability research now operates in language model settings. The certified radius of r = 1 in practice limits the formal claim's impact that the guarantee covers a narrow perturbation neighbourhood, and much of the observed improvement may stem from the empirical benefits of ensemble-style subsampling rather than the formal certificate per se. The relationship between the formal guarantee and the OOD generalisation improvement is argued intuitively but not formally established.

Originality

The key creative contribution is the identification of a clean analogy between input-space robustness certification (randomised smoothing for classifiers) and dataset-space stability certification for circuit discovery. Treating the concept dataset as the "input" subject to perturbation, and circuit component inclusion decisions as the "output" to be certified, is an elegant reframing. The adaptation of RS-Del to this setting is technically sound and well-executed.

---

> ### Author Rebuttal · Authors · 2026-03-31
>
> Dear reviewer CHgM,
>
> Thank you for the thorough and constructive review. We are glad you found the **theory well-grounded**, **OOD results impressive**, **the analogy elegant**, and the work **technically sound**. We appreciate the additional questions and suggestions, which we address below with new figures provided as PDF https://anonymous.4open.science/r/icml26-certified-circuits/rebuttal_new_figures.pdf.
>
> **Algorithm-agnosticism, Transformers, LLMs (W,Q2,Q4)**
>
> We performed new experiments for EAP-IG (edge attribution patching with integrated gradients, based on ACDC) on GPT2-Small, similar to the original paper, for 3 tasks: IOI, IOI Hard, and Greater-Than. Our certified circuits framework can be directly adapted from neurons (CNNs) to attention edges (LLMs), with the same certification algorithm. Findings are consistent, with certified EAP-IG matching or improving peak accuracy (up to +6pp) at smaller circuit sizes (up to 80% fewer edges) with full results in new Figures 1-3. Additionally, we show that these results extend to vision transformers (ViT-B/16), Figure 9. Including these new results, we now show **strong certified circuits performance across 3 architecture families (CNN, ViT, LLM) and 2 discovery algorithms (top-K, EAP-IG)**.
>
> **Ensemble effect vs. certification (W,Q3)**
>
> We directly compare the certification algorithm against a sampling baseline with majority voting, using the same masks at $p_{\text{del}}$ and number of samples n, without the $\tau$-threshold. New Figure 10 shows that certified circuits outperform simple ensembles in all ID and OOD settings, using the same compute runtime (new Table 2). **This confirms that specifically excluding neurons with statistically ambiguous votes (by abstaining) improves circuits beyond averaging**.
>
> **Larger radii (W,Q1)**
>
> The certified radius $r$ directly depends on $p_\text{del}$ and $\tau$. We add a new sweep across all 5 datasets for $r\in\\{1, 3, 11, 59\\}$ and $|D_c| \in \\{50, 100, 500, 1000\\}$. Even at $r=59$ with $|D_c|=1000$, certified circuits outperform the baseline on all circuit sizes, both ID and OOD. Extreme radii degrade the performance when the concept dataset size $|D_c|$ is small, because high $p_\text{del}$ masks most (and sometimes all) of the data. The LLM tasks (new Figure 1) naturally result in larger certified radii (cACC peaks at $r=59$ in new Fig. 1 for the task Greater-Than) due to larger concept datasets. **This shows that certified circuits empirically outperform the baseline while providing strong guarantees at larger radii.**
>
> **Abstention characterization (W)**
>
> We analyze several parameters, including abstention rates across radii, concept dataset sizes, and ID and OOD settings in new Figures 5 and 6, and the number of samples in Figure 13. Abstention rates are relatively stable between $25\%$ and $50\%$, correlated with increases in radius and negatively correlated with concept dataset size.
>
> **Formal link between certificate and OOD improvement (W)**
>
> We acknowledge this is a fair observation. While we do not have a formal proof that certification implies OOD generalization, three lines of evidence support the connection: (1) the majority-vote ablation (new Fig. 10) isolates $\tau$'s (i.e., abstentions') contribution beyond ensembling, (2) IoU analysis shows certified circuits converge to structurally consistent cores across distribution shifts (Figure 4, main paper), (3) neuron visualizations reveal qualitatively that abstained neurons encode generic or spurious features while certified neurons capture class-specific ones (new Figure 7, new Figures 16-17), and (4) certified circuits exhibit very high structural stability across different seeds (new Fig. 8). Formalizing this link, however, is a very intriguing direction for future work.
>
> **Additional analyses (prompted by other reviews)**
>
> We additionally provide uncertainty estimates (mean $\pm$ SE across $100$ classes) with significance tests in new Table 1; most cACC improvements are significant at $p < 0.001$.
>
> We also verify that the certification procedure itself converges: pairwise IoU across 40 random seeds yields a mean of $0.973$ even for the least stable classes in ImageNet-discovered certified circuits (new Figure 8). This is a strong finding since varying the seeds also varies the masked sampling, which addresses the circuit structure high-variance problem in the field.
>
> In summary, new experiments now span 3 architectures (CNN, ViT, LLM), 2 discovery algorithms (Top-K, EAP-IG), comprehensive ablations, and radius/abstention analysis, addressing the raised concerns and further strengthening the contribution. We thank the reviewer again for raising these points and will integrate all additions into the revised manuscript.

---

> > ### Author Rebuttal · Reviewer_CHgM · 2026-04-04
> >
> > I thank the authors for their rebuttal. The new experiments support the algorithm- and architecture-agnostic claims, hence I would like to maintain my favourable rating of the paper. However, for the lack of a formal link between certification and OOD improvement as acknowledged by the authors, I will not improve my score at this stage.

---

> > > ### Author Response · Authors · 2026-04-06
> > >
> > > We thank the reviewer for the engagement and for maintaining their favourable rating. We are pleased that the new experiments fully resolved the algorithm- and architecture-agnosticism concerns.
> > >
> > > On the remaining point, we would like to clarify the relationship between certification and OOD in our paper. Our core contribution is a **provable stability guarantee** for circuit discovery (Theorem 3.1), which is **formally proven** and now **empirically validated** across 3 architectures, 2 discovery algorithms, and 9 benchmarks, covering both ID and OOD settings. The OOD results are not a separate formal claim; they serve as empirical evidence that the certificate is practically meaningful: by removing neurons whose inclusion is unstable under dataset perturbations, certified circuits better isolate the target concept, which is why they transfer. We use OOD generalization to evaluate the quality of the discovered circuits.
> > >
> > > We agree with the reviewer that formalizing this connection is a compelling direction for future work. We note that an analogous gap exists in adversarial robustness, where the relationship between robustness certificates and OOD generalization remains an active research area with only partial and assumption-dependent theoretical results (Yi et al., 2021).
> > >
> > > We will clarify this distinction in the revised manuscript.
> > >
> > > Yi, Mingyang, et al. "Improved ood generalization via adversarial training and pretraing." ICML 2021.

---

### Decision · Program_Chairs · 2026-04-30

**Decision:**

Accept (regular)

**Comment:**

This paper develops a certification framework with randomized smoothing for circuit discovery. Reviewers agree that this is a solid paper. The new problem studied is original and important. The paper is well-written with sound theoretical results. Beyond certification, the paper also shows interesting results on improving OOD performance. A limitation is that the scope of the experiments is kind of limited. Overall, this paper can be accepted.